# EXPERIENCE-BASED KNOWLEDGE CORRECTION FOR ROBUST PLANNING IN MINECRAFT

**Seungjoon Lee[1], Suhwan Kim[1], Minhyeon Oh[1], Youngsik Yoon[1], Jungseul Ok[1,2][†]**
[1]Department of Computer Science and Engineering, POSTECH
[2]Graduate School of Artificial Intelligence, POSTECH
{sjlee1218, suhwankim, minhyeon.oh, ysyoon97, jungseul}@postech.ac.kr

## ABSTRACT

Large Language Model (LLM)-based planning has advanced embodied agents in long-horizon environments such as Minecraft, where acquiring latent knowledge of goal (or item) dependencies and feasible actions is critical. However, LLMs often begin with flawed priors and fail to correct them through prompting, even with feedback. We present XENON (eXpErience-based kNOwledge correctioN), an agent that algorithmically revises knowledge from experience, enabling robustness to flawed priors and sparse binary feedback. XENON integrates two mechanisms: Adaptive Dependency Graph, which corrects item dependencies using past successes, and Failure-aware Action Memory, which corrects action knowledge using past failures. Together, these components allow XENON to acquire complex dependencies despite limited guidance. Experiments across multiple Minecraft benchmarks show that XENON outperforms prior agents in both knowledge learning and long-horizon planning. Remarkably, with only a 7B open-weight LLM, XENON surpasses agents that rely on much larger proprietary models. Project page: https://sjlee-me.github.io/XENON

## 1 INTRODUCTION

Large Language Model (LLM)-based planning has advanced in developing embodied AI agents that tackle long-horizon goals in complex, real-world-like environments (Szot et al., 2021; Fan et al., 2022). Among such environments, Minecraft has emerged as a representative testbed for evaluating planning capability that captures the complexity of such environments (Wang et al., 2023b;c; Zhu et al., 2023; Yuan et al., 2023; Feng et al., 2024; Li et al., 2024b). Success in these environments often depends on agents acquiring planning knowledge, including the dependencies among goal items and the valid actions needed to obtain them. For instance, to obtain an iron nugget 🪙, an agent should first possess an iron ingot 🍪, which can only be obtained by the action *smelt*.

However, LLMs often begin with flawed priors about these dependencies and actions. This issue is indeed critical, since a lack of knowledge for a single goal can invalidate all subsequent plans that depend on it (Guss et al., 2019; Lin et al., 2021; Mao et al., 2022). We find several failure cases stemming from these flawed priors, a problem that is particularly pronounced for the lightweight LLMs suitable for practical embodied agents. First, an LLM often fails to predict planning knowledge accurately enough to generate a successful plan (Figure 1b), resulting in a complete halt in progress toward more challenging goals. Second, an LLM cannot robustly correct its flawed knowledge, even when prompted to self-correct with failure feedback (Shinn et al., 2023; Chen et al., 2024), often repeating the same errors (Figures 1c and 1d). To improve self-correction, one can employ more advanced techniques that leverage detailed reasons for failure (Zhang et al., 2024; Wang et al., 2023a). Nevertheless, LLMs often stubbornly adhere to their erroneous parametric knowledge (i.e. knowledge implicitly stored in model parameters), as evidenced by Stechly et al. (2024) and Du et al. (2024).

In response, we propose XENON (eXpErience-based kNOwledge correctioN), an agent that robustly learns planning knowledge from only binary success/failure feedback. To this end, instead of relying on an LLM for correction, XENON algorithmically and directly revises its external knowledge memory using its own experience, which in turn guides its planning. XENON learns this planning

---

[†]Corresponding author: Jungseul Ok <jungseul@postech.ac.kr>

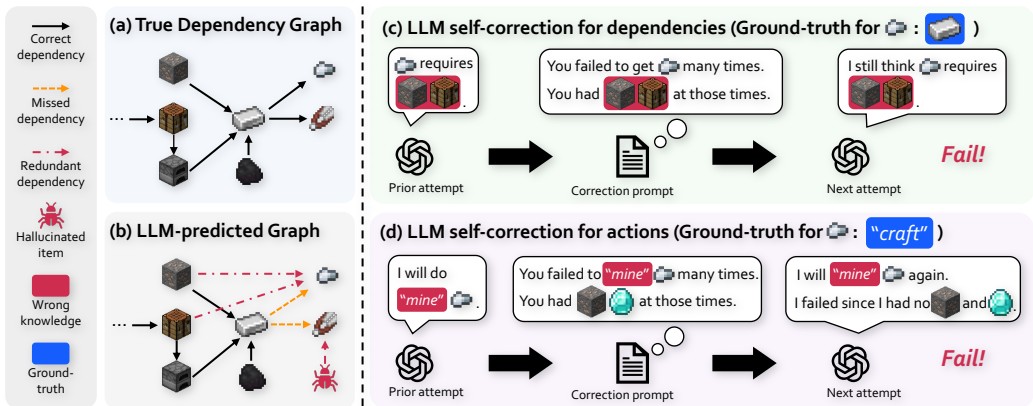

Figure 1: **An LLM exhibits flawed planning knowledge and fails at self-correction.** (b) The dependency graph predicted by Qwen2.5-VL-7B (Bai et al., 2025) contains multiple errors (e.g., missed dependencies, hallucinated items) compared to (a) the ground truth. (c, d) The LLM fails to correct its flawed knowledge about dependencies and actions from failure feedbacks, often repeating the same errors. See Appendix B for the full prompts and LLM's self-correction examples.

knowledge through two synergistic components. The first component, Adaptive Dependency Graph (ADG), revises flawed dependency knowledge by leveraging successful experiences to propose plausible new required items. The second component, Failure-aware Action Memory (FAM), builds and corrects its action knowledge by exploring actions upon failures. In the challenging yet practical setting of using only binary feedbacks, FAM enables XENON to disambiguate the cause of a failure, distinguishing between flawed dependency knowledge and invalid actions, which in turn triggers a revision in ADG for the former.

Extensive experiments in three Minecraft testbeds show that XENON excels at both knowledge acquisition and planning. XENON outperforms prior agents in learning knowledge, showing unique robustness to LLM hallucinations and modified ground-truth environmental rules. Furthermore, with only a 7B LLM, XENON significantly outperforms prior agents that rely on much larger proprietary models like GPT-4 in solving diverse long-horizon goals. These results suggest that robust algorithmic knowledge management can be a promising direction for developing practical embodied agents with lightweight LLMs (Belcak et al., 2025).

Our contributions are as follows. First, we propose XENON, an LLM-based agent that robustly learns planning knowledge from experience via algorithmic knowledge correction, instead of relying on the LLM to self-correct its own knowledge. We realize this idea through two synergistic mechanisms that explicitly store planning knowledge and correct it: Adaptive Dependency Graph (ADG) for correcting dependency knowledge based on successes, and Failure-aware Action Memory (FAM) for correcting action knowledge and disambiguating failure causes. Second, extensive experiments demonstrate that XENON significantly outperforms prior state-of-the-art agents in both knowledge learning and long-horizon goal planning in Minecraft.

## 2 RELATED WORK

### 2.1 LLM-BASED PLANNING IN MINECRAFT

Prior work has often address LLMs' flawed planning knowledge in Minecraft using impractical methods. For example, such methods typically involve directly injecting knowledge through LLM fine-tuning (Zhao et al., 2023; Feng et al., 2024; Liu et al., 2025; Qin et al., 2024) or relying on curated expert data (Wang et al., 2023c; Zhu et al., 2023; Wang et al., 2023a).

Another line of work attempts to learn planning knowledge via interaction, by storing the experience of obtaining goal items in an external knowledge memory. However, these approaches are often limited by unrealistic assumptions or lack robust mechanisms to correct the LLM's flawed prior knowledge. For example, ADAM and Optimus-1 artificially simplify the challenge of predicting and learning dependencies via shortcuts like pre-supplied items, while also relying on expert data such as learning curriculum (Yu & Lu, 2024) or Minecraft wiki (Li et al., 2024b). They also lack a robust

way to correct wrong action choices in a plan: ADAM has none, and Optimus-1 relies on unreliable LLM self-correction. Our most similar work, DECKARD (Nottingham et al., 2023), uses an LLM to predict item dependencies but does not revise its predictions for items that repeatedly fail, and when a plan fails, it cannot disambiguate whether the failure is due to incorrect dependencies or incorrect actions. In contrast, our work tackles the more practical challenge of learning planning knowledge and correcting flawed priors from only binary success/failure feedback.

## 2.2 LLM-BASED SELF-CORRECTION

LLM self-correction, i.e., having an LLM correct its own outputs, is a promising approach to overcome the limitations of flawed parametric knowledge. However, for complex tasks like planning, LLMs struggle to identify and correct their own errors without external feedback (Huang et al., 2024; Tyen et al., 2024). To improve self-correction, prior works fine-tune LLMs (Yang et al., 2025) or prompt LLMs to correct themselves using environmental feedback (Shinn et al., 2023) and tool-execution results (Gou et al., 2024). While we also use binary success/failure feedbacks, we directly correct the agent's knowledge in external memory by leveraging experience, rather than fine-tuning the LLM or prompting it to self-correct.

## 3 PRELIMINARIES

We aim to develop an agent capable of solving long-horizon goals by learning planning knowledge from experience. As a representative environment which necessitates accurate planning knowledge, we consider Minecraft as our testbed. Minecraft is characterized by strict dependencies among game items (Guss et al., 2019; Fan et al., 2022), which can be formally represented as a directed acyclic graph $\mathcal{G}^* = (\mathcal{V}^*, \mathcal{E}^*)$, where $\mathcal{V}^*$ is the set of all items and each edge $(u, q, v) \in \mathcal{E}^*$ indicates that $q$ quantities of an item $u$ are required to obtain an item $v$.[1] A goal is to obtain an item $g \in \mathcal{V}^*$. To obtain $g$, an agent must possess all of its prerequisites as defined by $\mathcal{G}^*$ in its inventory, and perform the valid high-level action in $\mathcal{A} = \{$"mine", "craft", "smelt"$\}$.

**Framework: Hierarchical agent with graph-augmented planning.** We employ a hierarchical agent with an LLM planner and a low-level controller, adopting a graph-augmented planning strategy (Li et al., 2024b; Nottingham et al., 2023). In this strategy, agent maintains its knowledge graph $\mathcal{G}$ and plans with $\mathcal{G}$ to decompose a goal $g$ into subgoals in two stages. First, the agent identifies prerequisite items it does not possess by traversing $\hat{\mathcal{G}}$ backward from $g$ to nodes with no incoming edges (i.e., basic items with no known requirements), and aggregates them into a list of (quantity, item) tuples, $((q_1, u_1), ..., (q_{L_g}, u_{L_g}) = (1, g))$. Second, the planner LLM converts this list into executable language subgoals $\{(a_l, q_l, u_l)\}_{l=1}^{L_g}$, where it takes each $u_l$ as input and outputs a high-level action $a_l$ to obtain $u_l$. Then the controller executes each subgoal, i.e., it takes each language subgoal as input and outputs a sequence of low-level actions in the environment to achieve it. After each subgoal execution, the agent receives only binary success/failure feedback.

**Problem formulation: Dependency and action learning.** To plan correctly, the agent must acquire knowledge of the true dependency graph $\mathcal{G}^*$. However, $\mathcal{G}^*$ is latent, making it necessary for the agent to learn this structure from experience. We model this as revising a learned graph, $\hat{\mathcal{G}} = (\hat{\mathcal{V}}, \hat{\mathcal{E}})$, where $\hat{\mathcal{V}}$ contains known items and $\hat{\mathcal{E}}$ represents the agent's current belief about item dependencies. Following Nottingham et al. (2023), whenever the agent obtains a new item $v$, it identifies the *experienced requirement set* $\mathcal{R}_{\exp}(v)$, the set of (item, quantity) pairs consumed during this item acquisition. The agent then updates $\hat{\mathcal{G}}$ by replacing all existing incoming edges to $v$ with the newly observed $\mathcal{R}_{\exp}(v)$. The detailed update procedure is in Appendix C.

We aim to maximize the accuracy of learned graph $\hat{\mathcal{G}}$ against true graph $\mathcal{G}^*$. We define this accuracy $N_{true}(\hat{\mathcal{G}})$ as the number of items whose incoming edges are identical in $\hat{\mathcal{G}}$ and $\mathcal{G}^*$, i.e.,

$$N_{true}(\hat{\mathcal{G}}) := \sum_{v \in \mathcal{V}^*} \mathbb{I}(\mathcal{R}(v, \hat{\mathcal{G}}) = \mathcal{R}(v, \mathcal{G}^*)) , \qquad (1)$$

---

[1]In our actual implementation, each edge also stores the resulting item quantity, but we omit it from the notation for presentation simplicity, since most edges have resulting item quantity 1 and this multiplicity is not essential for learning item dependencies.

where the dependency set, $\mathcal{R}(v, \mathcal{G})$, denotes the set of all incoming edges to the item $v$ in the graph $\mathcal{G}$.

# 4 METHODS

XENON is an LLM-based agent with two core components: Adaptive Dependency Graph (ADG) and Failure-aware Action Memory (FAM), as shown in Figure 3. ADG manages dependency knowledge, while FAM manages action knowledge. The agent learns this knowledge in a loop that starts by selecting an unobtained item as an exploratory goal (detailed in Appendix G). Once an item goal $g$ is selected, ADG, our learned dependency graph $\mathcal{G}$, traverses itself to construct $((q_1, u_1), \ldots, (q_{L_g}, u_{L_g}) = (1, g))$. For each $u_l$ in this list, FAM either reuses a previously successful action for $u_l$ or, if none exists, the planner LLM selects a high-level action $a_l \in \mathcal{A}$ given $u_l$ and action histories from FAM. The resulting actions form language subgoals $\{(a_l, q_l, u_l)\}_{l=1}^{L_g}$. The controller then takes each subgoal as input, executes a sequence of low-level actions to achieve it, and returns binary success/failure feedback, which is used to update both ADG and FAM. The full procedure is outlined in Algorithm 1 in Appendix D. We next detail each component, beginning with ADG.

## 4.1 ADAPTIVE DEPENDENCY GRAPH (ADG)

**Dependency graph initialization.** To make the most of the LLM's prior knowledge, albeit incomplete, we initialize the learned dependency graph $\hat{\mathcal{G}} = (\hat{\mathcal{V}}, \hat{\mathcal{E}})$ using an LLM. We follow the initialization process of DECKARD (Nottingham et al., 2023), which consists of two steps. First, $\hat{\mathcal{V}}$ is assigned $\mathcal{V}_0$, which is the set of goal items whose dependencies must be learned, and $\hat{\mathcal{E}}$ is assigned $\emptyset$. Second, for each item $v$ in $\hat{\mathcal{V}}$, the LLM is prompted to predict its requirement set (i.e. incoming edges of $v$), aggregating them to construct the initial graph.

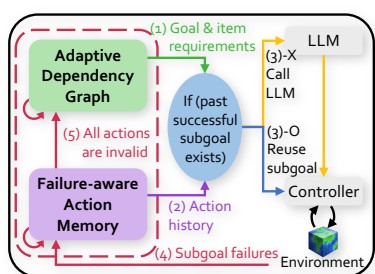

Figure 3: Overview. XENON updates Adaptive Dependency Graph and Failure-aware Action Memory with environmental experiences.

However, those LLM-predicted requirement sets often include items not present in the initial set $\mathcal{V}_0$, which is a phenomenon overlooked by DECKARD. Since $\mathcal{V}_0$ may be an incomplete subset of all possible game items $\mathcal{V}^*$, we cannot determine whether such items are genuine required items or hallucinated items which do not exist in the environment. To address this, we provisionally accept all LLM requirement set predictions. We iteratively expand the graph by adding any newly mentioned item to $\hat{\mathcal{V}}$ and, in turn, querying the LLM for its own requirement set. This expansion continues until a requirement set has been predicted for every item in $\hat{\mathcal{V}}$. Since we assume that the true graph $\mathcal{G}^*$ is a DAG, we algorithmically prevent cycles in $\hat{\mathcal{G}}$; see Appendix E.2 for the cycle-check procedure. The quality of this initial LLM-predicted graph is analyzed in detail in Appendix K.1.

**Dependency graph revision.** Correcting the agent's flawed dependency knowledge involves two challenges: (1) detecting and handling hallucinated items from the graph initialization, and (2) proposing a new requirement set. Simply prompting an LLM for corrections is ineffective, as it often predicts a new, flawed requirement set, as shown in Figures 1c and 1d. Therefore, we revise $\hat{\mathcal{G}}$ algorithmically using the agent's experiences, without relying on the LLM.

To implement this, we introduce a dependency revision procedure called `RevisionByAnalogy` and a revision count $C(v)$ for each item $v \in \hat{\mathcal{V}}$. This procedure outputs a revised graph by taking item $v$ whose dependency needs to be revised, its revision count $C(v)$, and the current graph $\hat{\mathcal{G}}$ as inputs, leveraging the required items of previously obtained items. When a revision for an item $v$ is triggered by FAM (Section 4.2), the procedure first discards $v$'s existing requirement set (i.e., $\mathcal{R}(v, \hat{\mathcal{G}}) \leftarrow \emptyset$). It increments the revision count $C(v)$ for $v$. Based on whether $C(v)$ exceeds a hyperparameter $c_0$, `RevisionByAnalogy` proceeds with one of the following two cases:

- **Case 1: Handling potentially hallucinated items ($C(v) > c_0$).** If an item $v$ remains unobtainable after excessive revisions, the procedure flags it as *inadmissible* to signify that it may be a hallucinated item. This reveals a critical problem: if $v$ is indeed a hallucinated item, any of its descendants

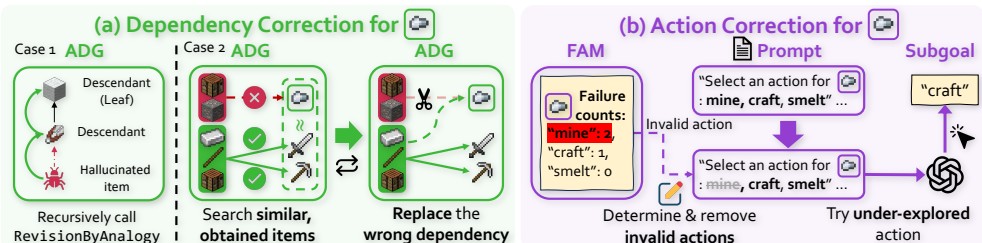

Figure 4: XENON's algorithmic knowledge correction. (a) Dependency Correction via `RevisionByAnalogy`. Case 1: For an inadmissible item (e.g., a hallucinated item), its descendants are recursively revised to remove the flawed dependency. Case 2: A flawed requirement set is revised by referencing similar, obtained items. (b) Action Correction via FAM. FAM prunes invalid actions from the LLM's prompt based on failures, guiding it to select an under-explored action.

in $\hat{\mathcal{G}}$ become permanently unobtainable. To enable XENON to try these descendant items through alternative paths, we recursively call `RevisionByAnalogy` for all of $v$'s descendants in $\hat{\mathcal{G}}$, removing their dependency on the inadmissible item $v$ (Figure 4a, Case 1). Finally, to account for cases where $v$ may be a genuine item that is simply difficult to obtain, its requirement set $\mathcal{R}(v, \hat{\mathcal{G}})$ is reset to a general set of all resource items (i.e. items previously consumed for crafting other items), each with a quantity of hyperparameter $\alpha_i$.

- **Case 2: Plausible revision for less-tried items ($C(v) \leq c_0$).** The item $v$'s requirement set, $\mathcal{R}(v, \hat{\mathcal{G}})$, is revised to determine both a plausible set of new items and their quantities. First, for plausible required items, we use an idea that similar goals often share similar preconditions (Yoon et al., 2024). Therefore, we set the new required items referencing the required items of the top-$K$ similar, successfully obtained items (Figure 4a, Case 2). We compute this item similarity as the cosine similarity between the Sentence-BERT (Reimers & Gurevych, 2019) embeddings of item names. Second, to determine their quantities, the agent should address the trade-off between sufficient amounts to avoid failures and an imperfect controller's difficulty in acquiring them. Therefore, the quantities of those new required items are determined by gradually scaling with the revision count, $\alpha_s C(v)$.

Here, the hyperparameter $c_0$ serves as the revision count threshold for flagging an item as inadmissible. $\alpha_i$ and $\alpha_s$ control the quantity of each required item for inadmissible items (Case 1), and for less-tried items (Case 2), respectively, to maintain robustness when dealing with an imperfect controller. $K$ determines the number of similar, successfully obtained items to reference for (Case 2). Detailed pseudocode of RevisionByAnalogy is in Appendix E.3, Algorithm 3.

## 4.2 FAILURE-AWARE ACTION MEMORY (FAM)

FAM is designed to address two challenges of learning only from binary success/failure feedback: (1) discovering valid high-level actions for each item, and (2) disambiguating the cause of persistent failures between invalid actions and flawed dependency knowledge. This section first describes FAM's core mechanism, and then details how it addresses each of these challenges in turn.

**Core mechanism: empirical action classification.** FAM classifies actions as either *empirically valid* or *empirically invalid* for each item, based on their history of past subgoal outcomes. Specifically, for each item $v \in \hat{\mathcal{V}}$ and action $a \in \mathcal{A}$, FAM maintains the number of successful and failed outcomes, denoted as $S(a, v)$ and $F(a, v)$ respectively. Based on these counts, an action $a$ is classified as *empirically invalid* for $v$ if it has failed repeatedly, (i.e., $F(a, v) \geq S(a, v) + x_0$); otherwise, it is classified as *empirically valid* if it has succeeded at least once (i.e., $S(a, v) > 0$ and $S(a, v) > F(a, v) - x_0$). The hyperparameter $x_0$ controls the tolerance for this classification, accounting for the possibility that an imperfect controller might fail even with an indeed valid action.

**Addressing challenge 1: discovering valid actions.** FAM helps XENON discover valid actions by avoiding repeatedly failed actions when making a subgoal $sg_l = (a_l, q_l, u_l)$. Only when FAM has no empirically valid action for $u_l$, XENON queries the LLM to select an under-explored action for constructing $sg_l$. To accelerate this search for a valid action, we query the LLM with (i) the

current subgoal item $u_l$, (ii) empirically valid actions for top-$K$ similar items successfully obtained and stored in FAM (using Sentence-BERT similarity as in Section 4.1), and (iii) candidate actions for $u_l$ that remain after removing all empirically invalid actions from $\mathcal{A}$ (Figure 4b). We prune action candidates rather than include the full failure history because LLMs struggle to effectively utilize long prompts (Li et al., 2024a; Liu et al., 2024). If FAM already has an empirically valid one, XENON reuses it to make $sg_l$ without using LLM. Detailed procedures and prompts are in Appendix F.

**Addressing challenge 2: disambiguating failure causes.** By ensuring systematic action exploration, FAM allows XENON to determine that persistent subgoal failures stem from flawed dependency knowledge rather than from the actions. Specifically, once FAM classifies all actions in $\mathcal{A}$ for an item as empirically invalid, XENON concludes that the error lies within ADG and triggers its revision. Subsequently, XENON resets the item's history in FAM to allow for a fresh exploration of actions with the revised ADG.

### 4.3 ADDITIONAL TECHNIQUE: CONTEXT-AWARE REPROMPTING (CRE) FOR CONTROLLER

In real-world-like environments, an imperfect controller can stall (e.g., in deep water). To address this, XENON employs context-aware reprompting (CRe), where an LLM uses the current image observation and the controller's language subgoal to decide whether to replace the subgoal and propose a new temporary subgoal to escape the stalled state (e.g., "get out of the water"). Our CRe is adapted from Optimus-1 (Li et al., 2024b) to be suitable for smaller LLMs, with two differences: (1) a two-stage reasoning process that captions the observation first and then makes a text-only decision on whether to replace the subgoal, and (2) a conditional trigger that activates only when the subgoal for item acquisition makes no progress, rather than at fixed intervals. See Appendix H for details.

## 5 EXPERIMENTS

### 5.1 SETUPS

**Environments.** We conduct experiments in three Minecraft environments, which we separate into two categories based on their controller capacity. First, as realistic, visually-rich embodied AI environments, we use MineRL (Guss et al., 2019) and Mineflayer (PrismarineJS, 2023) with imperfect low-level controllers: STEVE-1 (Lifshitz et al., 2023) in MineRL and hand-crafted codes (Yu & Lu, 2024) in Mineflayer. Second, we use MC-TextWorld (Zheng et al., 2025) as a controlled testbed with a perfect controller. Each experiment in this environment is repeated over 15 runs; in our results, we report the mean and standard deviation, omitting the latter when it is negligible. In all environments, the agent starts with an empty inventory. Further details on environments are provided in Appendix J. Additional experiments in a household task planning domain other than Minecraft are reported in Appendix A, where XENON also exhibits robust performance.

**Evaluation metrics.** For both dependency learning and planning evaluations, we utilize the 67 goals from 7 groups proposed in the long-horizon task benchmark (Li et al., 2024b). To evaluate dependency learning with an intuitive performance score between 0 and 1, we report $N_{\text{true}}(\hat{\mathcal{G}})/67$, where $N_{\text{true}}(\hat{\mathcal{G}})$ is defined in Equation (1). We refer to this normalized score as Experienced Graph Accuracy (EGA). To evaluate planning performance, we follow the benchmark setting (Li et al., 2024b): at the beginning of each episode, a goal item is specified externally for the agent, and we measure the average success rate (SR) of obtaining this goal item in MineRL. See Table 10 for the full list of goals.

Table 1: Comparison of knowledge correction mechanisms across agents. ○: Our proposed mechanism (XENON), △: LLM self-correction, ✗: No correction, –: Not applicable.

| Agent | Dependency Correction | Action Correction |
|---|---|---|
| XENON | ○ | ○ |
| SC | △ | △ |
| DECKARD | ✗ | ✗ |
| ADAM | - | ✗ |
| RAND | ✗ | ✗ |

**Implementation details.** For the planner, we use Qwen2.5-VL-7B (Bai et al., 2025). The learned dependency graph is initialized with human-written plans for three goals ("craft an iron sword 🗡", "craft a golden sword 🗡," "mine a diamond 💎"), providing minimal knowledge; the agent must learn dependencies for over 80% of goal items through experience. We employ CRe only for long-horizon

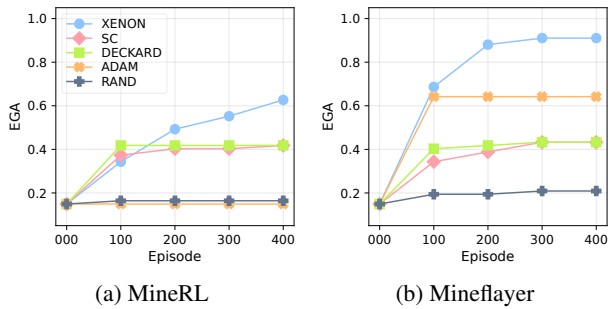

(a) MineRL  (b) Mineflayer

Figure 5: **Robustness against flawed prior knowledge.**
EGA over 400 episodes in (a) MineRL and (b) Mineflayer.
XENON consistently outperforms the baselines.

Table 2: **Robustness to LLM hallucinations.** The number of correctly learned dependencies of items that are descendants of a hallucinated item in the initial LLM-predicted dependency graph (out of 12).

| Agent | Learned descendants of hallucinated items |
|---|---|
| XENON | **0.33** |
| SC | 0 |
| ADAM | 0 |
| DECKARD | 0 |
| RAND | 0 |

goal planning in MineRL. All hyperparameters are kept consistent across experiments. Further details on hyperparameters and human-written plans are in Appendix I.

**Baselines.** As no prior work learns dependencies in our exact setting, we adapt four baselines, whose knowledge correction mechanisms are summarized in Table 1. For dependency knowledge, (1) **LLM Self-Correction (SC)** starts with an LLM-predicted dependency graph and prompts the LLM to revise it upon failures; (2) **DECKARD** (Nottingham et al., 2023) also relies on an LLM-predicted graph but with no correction mechanism; (3) **ADAM** (Yu & Lu, 2024) assumes that any goal item requires all previously used resource items, each in a sufficient quantity; and (4) **RAND**, the simplest baseline, uses a static graph similar to DECKARD. Regarding action knowledge, all baselines except for RAND store successful actions. However, only the SC baseline attempts to correct its flawed knowledge upon failures. The SC prompts the LLM to revise both its dependency and action knowledge using previous LLM predictions and interaction trajectories, as done in many self-correction methods (Shinn et al., 2023; Stechly et al., 2024). See Appendix B for the prompts of SC and Appendix J.1 for detailed descriptions of these baselines. To evaluate planning on diverse long-horizon goals, we further compare XENON with recent planning agents that are provided with oracle dependencies: DEPS Wang et al. (2023b), Jarvis-1 Wang et al. (2023c), Optimus-1 Li et al. (2024b), and Optimus-2 Li et al. (2025b).

## 5.2 ROBUST DEPENDENCY LEARNING AGAINST FLAWED PRIOR KNOWLEDGE

XENON demonstrates robust dependency learning from flawed prior knowledge, consistently outperforming baselines with an EGA of approximately 0.6 in MineRL and 0.9 in Mineflayer (Figure 5), despite the challenging setting with imperfect controllers. This superior performance is driven by its algorithmic correction mechanism, `RevisionByAnalogy`, which corrects flawed dependency knowledge while also accommodating imperfect controllers by gradually scaling required items quantities. The robustness of this algorithmic correction is particularly evident in two key analyses of the learned graph for each agent from the MineRL experiments. First, as shown in Table 2, XENON is uniquely robust to LLM hallucinations, learning dependencies for descendant items of non-existent, hallucinated items in the initial LLM-predicted graph. Second, XENON outperforms the baselines in learning dependencies for items that are unobtainable by the initial graph, as shown in Table 13.

Our results demonstrate the unreliability of relying on LLM self-correction or blindly trusting an LLM's flawed knowledge; in practice, SC achieves the same EGA as DECKARD, with both plateauing around 0.4 in both environments.

We observe that controller capacity strongly impacts dependency learning. This is evident in ADAM, whose EGA differs markedly between MineRL ($\approx 0.1$), which has a limited controller, and Mineflayer ($\approx 0.6$), which has a more competent controller. While ADAM unrealistically assumes a controller can gather large quantities of all resource items before attempting a new item, MineRL's controller STEVE-1 (Lifshitz et al., 2023) cannot execute this demanding strategy, causing ADAM's EGA to fall below even the simplest baseline, RAND. Controller capacity also accounts for XENON's lower EGA in MineRL. For instance, XENON learns none of the dependencies of the Redstone group items, as STEVE-1 cannot execute XENON's strategy for *inadmissible items* (Section 4.1). In contrast, the

Table 3: **Performance on long-horizon task benchmark.** Average success rate of each group on the long-horizon task benchmark Li et al. (2024b) in MineRL. *Oracle* indicates that the true dependency graph is known in advance, *Learned* indicates that the graph is learned via experience across 400 episodes. For fair comparison across LLMs, we include Optimus-1[†], our reproduction of Optimus-1 using Qwen2.5-VL-7B. Due to resource limits, results for DEPS, Jarvis-1, Optimus-1, and Optimus-2 are cited directly from (Li et al., 2025b). See Appendix K.12 for the success rate on each goal.

| Method | Dependency | Planner LLM | Overall | Wood | Stone | Iron | Diamond | Gold | Armor | Redstone |
|--------|-----------|-------------|---------|------|-------|------|---------|------|-------|----------|
| DEPS | - | Codex | 0.22 | 0.77 | 0.48 | 0.16 | 0.01 | 0.00 | 0.10 | 0.00 |
| Jarvis-1 | Oracle | GPT-4 | 0.38 | 0.93 | 0.89 | 0.36 | 0.08 | 0.07 | 0.15 | 0.16 |
| Optimus-1 | Oracle | GPT-4V | 0.43 | 0.98 | 0.92 | 0.46 | 0.11 | 0.08 | 0.19 | 0.25 |
| Optimus-2 | Oracle | GPT-4V | 0.45 | **0.99** | **0.93** | 0.53 | 0.13 | 0.09 | 0.21 | 0.28 |
| Optimus-1[†] | Oracle | Qwen2.5-VL-7B | 0.34 | 0.92 | 0.80 | 0.22 | 0.10 | 0.09 | 0.17 | 0.04 |
| XENON * | Oracle | Qwen2.5-VL-7B | **0.79** | 0.95 | **0.93** | **0.83** | **0.75** | 0.73 | **0.61** | **0.75** |
| XENON | *Learned* | Qwen2.5-VL-7B | 0.54 | 0.85 | 0.81 | 0.46 | 0.64 | **0.74** | 0.28 | 0.00 |

more capable Mineflayer controller executes this strategy successfully, allowing XENON to learn the correct dependencies for 5 of 6 Redstone items. This difference highlights the critical role of controllers for dependency learning, as detailed in our analysis in Appendix K.3

## 5.3 EFFECTIVE PLANNING TO SOLVE DIVERSE GOALS

As shown in Table 3, XENON significantly outperforms baselines in solving diverse long-horizon goals despite using the lightweight Qwen2.5-VL-7B LLM (Bai et al., 2025), while the baselines rely on large proprietary models such as Codex (Chen et al., 2021), GPT-4 (OpenAI, 2024), and GPT-4V (OpenAI, 2023). Remarkably, even with its *learned* dependency knowledge (Section 5.2), XENON surpasses the baselines with the oracle knowledge on challenging late-game goals, achieving high SRs for item groups like Gold (0.74) and Diamond (0.64).

XENON's superiority stems from two key factors. First, its FAM provides systematic, fine-grained action correction for each goal. Second, it reduces reliance on the LLM for planning in two ways: it shortens prompts and outputs by requiring it to predict one action per subgoal item, and it bypasses the LLM entirely by reusing successful actions from FAM. In contrast, the baselines lack a systematic, fine-grained action correction mechanism and instead make LLMs generate long plans from lengthy prompts—a strategy known to be ineffective for LLMs (Wu et al., 2024; Li et al., 2024a). This challenge is exemplified by Optimus-1[†]. Despite using a knowledge graph for planning like XENON, its long-context generation strategy causes LLM to predict incorrect actions or omit items explicitly provided in its prompt, as detailed in Appendix K.5.

We find that accurate knowledge is critical for long-horizon planning, as its absence can make even a capable agent ineffective. The Redstone group from Table 3 provides an example: while XENON* with oracle knowledge succeeds (0.75 SR), XENON with learned knowledge fails entirely (0.00 SR), because it failed to learn the dependencies for Redstone goals due to the controller's limited capacity in MineRL (Section 5.2). This finding is further supported by our comprehensive ablation study, which confirms that accurate dependency knowledge is most critical for success across all goals (See Table 17 in Appendix K.7).

## 5.4 ROBUST DEPENDENCY LEARNING AGAINST KNOWLEDGE CONFLICTS

To isolate dependency learning from controller capacity, we shift to the MC-TextWorld environment with a perfect controller. In this setting, we test each agent's robustness to conflicts with its prior knowledge (derived from the LLM's initial predictions and human-written plans) by introducing arbitrary perturbations to the ground-truth required items and actions. These perturbations are applied with an intensity level; a higher intensity affects a greater number of items, as shown in Table 4. This intensity is denoted by a tuple

Table 4: Effect of ground-truth perturbations on prior knowledge.

| Perturbation Intensity | Goal items obtainable via prior knowledge |
|------------------------|-------------------------------------------|
| 0 | 16 (no perturbation) |
| 1 | 14 (12 %) |
| 2 | 11 (31 %) |
| 3 | 9 (44 %) |

`(r,a)` for required items and actions, respectively. `(0,0)` represents the vanilla setting with no perturbations. See Figure 21 for the detailed perturbation process.

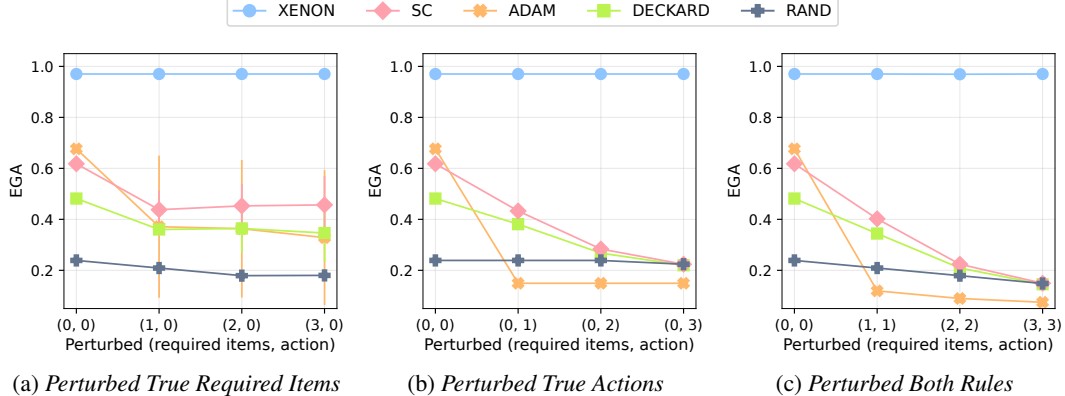

(a) *Perturbed True Required Items*     (b) *Perturbed True Actions*     (c) *Perturbed Both Rules*

Figure 6: **Robustness against knowledge conflicts.** EGA after 3,000 environment steps in MC-TextWorld under different perturbations of the ground-truth rules. The plots show performance with increasing intensities of perturbation applied to: (a) requirements only, (b) actions only, and (c) both (see Table 4).

Figure 6 shows XENON's robustness to knowledge conflicts, as it maintains a near-perfect EGA (≈0.97). In contrast, the performance of all baselines degrades as perturbation intensity increases across all three perturbation scenarios (required items, actions, or both). We find that prompting an LLM to self-correct is ineffective when the ground truth conflicts with its parametric knowledge: SC shows no significant advantage over DECKARD, which lacks a correction mechanism. ADAM is vulnerable to action perturbations; its strategy of gathering all resource items before attempting a new item fails when the valid actions for those resources are perturbed, effectively halting its learning.

## 5.5 ABLATION STUDIES ON KNOWLEDGE CORRECTION MECHANISMS

As shown in Table 5, to analyze XENON's knowledge correction mechanisms for dependencies and actions, we conduct ablation studies in MC-TextWorld. While dependency correction is generally more important for overall performance, action correction becomes vital under action perturbations. In contrast, LLM self-correction is ineffective for complex scenarios: it offers minimal gains for dependency correction even in the vanilla setting and fails entirely for perturbed actions. Its effectiveness is limited to simpler scenarios, such as action correction in the vanilla setting. These results demonstrate that our algorithmic knowledge correction approach enables robust learning from experience, overcoming the limitations of both LLM self-correction and flawed initial knowledge.

Table 5: Ablation study of knowledge correction mechanisms. ○: XENON; △: LLM self-correction; ✗: No correction. All entries denote the EGA after 3,000 environment steps. Columns denote the perturbation setting (r,a). For LLM self-correction, we use the same prompt as the SC baseline (see Appendix B).

| Dependency Correction | Action Correction | (0,0) | (3,0) | (0,3) | (3,3) |
|:---:|:---:|:---:|:---:|:---:|:---:|
| ○ | ○ | 0.97 | 0.97 | 0.97 | 0.97 |
| ○ | △ | 0.93 | 0.93 | 0.12 | 0.12 |
| ○ | ✗ | 0.84 | 0.84 | 0.12 | 0.12 |
| △ | ○ | 0.57 | 0.30 | 0.57 | 0.29 |
| ✗ | ○ | 0.53 | 0.13 | 0.53 | 0.13 |
| ✗ | ✗ | 0.46 | 0.13 | 0.19 | 0.11 |

## 5.6 ABLATION STUDIES ON HYPERPARAMETERS

To validate XENON's stability to its hyperparameters, we conduct comprehensive ablation studies in both MC-TextWorld and MineRL. In these studies, we vary one hyperparameter at a time while keeping the others fixed to their default values ($c_0 = 3$, $\alpha_i = 8$, $\alpha_s = 2$, $x_0 = 2$).

Our results (Figure 7, Figure 8) show that although XENON is generally stable across hyperparameters, an effective learning strategy should account for controller capacity when the controller is imperfect. In MC-TextWorld (Figure 7), XENON maintains near-perfect EGA across a wide range of all tested hyperparameters, confirming its stability when a perfect controller is used. In MineRL (Figure 8), with an imperfect controller, the results demonstrate two findings. First, while influenced by hyperparameters, XENON still demonstrates robust performance, showing EGA after

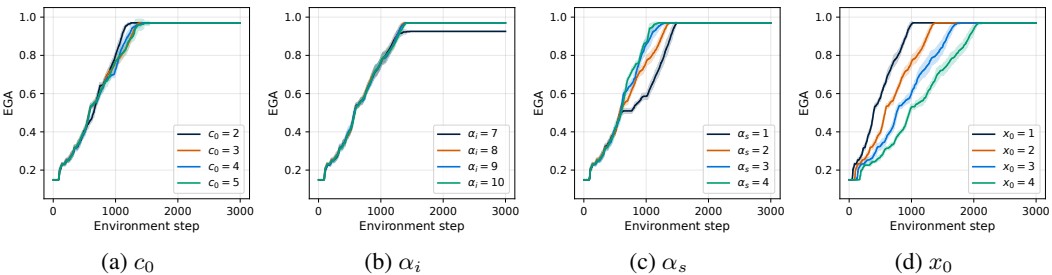

(a) $c_0$          (b) $\alpha_i$          (c) $\alpha_s$          (d) $x_0$

Figure 7: **Hyperparameter ablation study in MC-TextWorld.** EGA over 3,000 environment steps under different hyperparameters. The plots show EGA when varying: (a) $c_0$ (revision count threshold for inadmissible items), (b) $\alpha_i$ (required items quantities for inadmissible items), (c) $\alpha_s$ (required items quantities for less-tried items), and (d) $x_0$ (invalid action threshold). Each study varies one hyperparameter while keeping the others fixed to their default values ($c_0 = 3$, $\alpha_i = 8$, $\alpha_s = 2$, $x_0 = 2$).

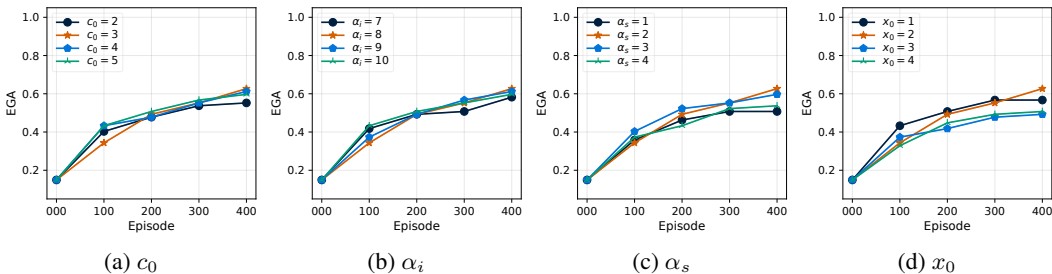

(a) $c_0$          (b) $\alpha_i$          (c) $\alpha_s$          (d) $x_0$

Figure 8: **Hyperparameter ablation study in MineRL.** EGA over 400 episodes under different hyperparameters. The plots show EGA when varying: (a) $c_0$ (revision count threshold for inadmissible items), (b) $\alpha_i$ (required items quantities for inadmissible items), (c) $\alpha_s$ (required items quantities for less-tried items), and (d) $x_0$ (invalid action threshold). Each study varies one hyperparameter while keeping the others fixed to their default values ($c_0 = 3, \alpha_i = 8, \alpha_s = 2, x_0 = 2$).

400 episodes for all tested values remains near or above 0.5, outperforming baselines that plateau around or below 0.4 (Figure 5a). Second, controller capacity should be considered when designing dependency and action learning strategies. For example, the ablation on $\alpha_s$ (Figure 8c) shows that while gathering a sufficient quantity of items is necessary ($\alpha_s = 1$), overburdening the controller with excessive items ($\alpha_s = 4$) also degrades performance. Similarly, the ablation on $x_0$ (Figure 8d) shows the need to balance tolerating controller failures against wasting time on invalid actions.

We provide additional ablations in the Appendix on dependency and action learning—when initializing the dependency graph from an external source mismatched to the environment (Figure 23), when scaling to more goals/actions (Figure 24), and when using a smaller 4B planner LLM (Figure 26)—as well as an ablation of action selection methods for subgoal construction (Figure 25).

## 6 CONCLUSION

We address the challenge of robust planning via experience-based algorithmic knowledge correction. With XENON, we show that directly revising external knowledge through experience enables an LLM-based agent to overcome flawed priors and sparse feedback, surpassing the limits of LLM self-correction. Experiments across diverse Minecraft benchmarks demonstrate that this approach not only strengthens knowledge acquisition and long-horizon planning, but also enables an agent with a lightweight 7B open-weight LLM to outperform prior methods that rely on much larger proprietary models. Our work delivers a key lesson for building robust LLM-based embodied agents: LLM priors should be treated with skepticism and continuously managed and corrected algorithmically.

**Limitations.** Despite its contributions, XENON faces a limitation. XENON's performance is influenced by the underlying controller; in MineRL, STEVE-1 (Lifshitz et al., 2023) controller struggles with spatial exploration tasks, making a performance gap compared to more competent controllers like Mineflayer. Future work could involve jointly training the planner and controller, potentially using hierarchical reinforcement learning.

ACKNOWLEDGMENTS

This work was supported by the Institute of Information & Communications Technology Planning & Evaluation (IITP) and IITP-ITRC (Information Technology Research Center) grant funded by the Korea government (MSIT) (No. RS-2019-II191906, Artificial Intelligence Graduate School Program (POSTECH); IITP-2026-RS-2024-00437866; RS-2024-00509258, Global AI Frontier Lab), by a grant from the Korea Institute for Advancement of Technology (KIAT), funded by the Ministry of Trade, Industry and Energy (MOTIE), Republic of Korea (RS-2025-00564342), and by Seoul R&BD Program (SP240008) through the Seoul Business Agency (SBA) funded by The Seoul Metropolitan Government.

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

This appendix is organized as follows:

- **Appendix A**: Experiments in a domain other than Minecraft (Microsoft TextWorld Cooking).
- **Appendix B**: Prompts and qualitative results of LLM self-correction in our experiments.
- **Appendix C**: Detailed procedure for experienced requirement set determination and dependency graph updates, as discussed in Section 3.
- **Appendix E**: Detailed pseudocode and the prompt for ADG in Section 4.1.
- **Appendix F**: Detailed pseudocode and the prompt for step-by-step planning using FAM in Section 4.2.
- **Appendix H**: Detailed descriptions and the prompt for CRe in Section 4.3.
- **Appendix I**: Detailed descriptions of implementation, human-written plans, and hyperparameters.
- **Appendix J**: Detailed descriptions of the baselines and experimental environments in Section 5.
- **Appendix K**: Analysis of experimental results and additional experimental results.
- **Appendix L**: Descriptions about LLM usage.

Table 6: Success rates in the TextWorld Cooking environment, comparing XENON against the SC (LLM self-correction) and DECKARD baselines from Section 5.1. We report the mean $\pm$ standard deviation over 3 independent runs, where each run consists of 100 episodes.

|  | DECKARD | SC | XENON |
|---|---|---|---|
| Success Rate | $0.09 \pm 0.02$ | $0.75 \pm 0.04$ | $1.00 \pm 0.00$ |

## A  ADDITIONAL EXPERIMENTS IN ANOTHER DOMAIN

To assess generalization beyond Minecraft, we evaluate XENON on the Microsoft TextWorld Cooking environment (Côté et al., 2018), a text-based household task planning benchmark. We demonstrate XENON can correct an LLM's flawed knowledge of preconditions (e.g., required tools) and valid actions for plans using ADG and FAM in this domain as well. We note that XENON is applied with minimal modification: FAM is applied without modification, while ADG is adapted from its original design, which supports multiple incoming edges (preconditions) for a node, to one that allows only a single incoming edge, as this domain requires only a single precondition per node.

### A.1  EXPERIMENT SETUP

**Environment Rules.** The goal is to prepare and eat a meal by reading a cookbook, which provides a plan as a list of (action, ingredient) pairs, e.g., ("fry", "pepper"). We note that an agent cannot succeed by naively following this plan. This is because the agent must solve two key challenges: (1) it must discover the valid **tool** required for each cookbook action, and (2) it must discover the valid, **executable action** for each cookbook action, as some cookbook actions are not directly accepted by the environment (i.e., not in its action space).

Specifically, to succeed a cookbook's (action, ingredient) pair, an agent must make a subgoal, formatted as (`executable action`, ingredient, `tool`), where the `executable action` and `tool` must be valid for the cookbook action. For example, the cookbook's ("fry", "pepper") pair requires the agent to make a subgoal (`cook`, "pepper", `stove`). The available executable action space consists of { "chop", "close", "cook", "dice", "drop", "eat", "examine", "slice", "prepare" }, and the available tools are { "knife", "oven", "stove", "fridge", "table", "counter" }.

**Baselines and Evaluation.** All agents use an LLM (Qwen2.5-VL-7B) to make subgoals. The `tool` for each cookbook action is predicted by the LLM from the available tools before an episode begins. At each timestep during the episode, given a cookbook action, the LLM predicts an `executable action` from the executable action space, constructing a subgoal from this predicted `executable action`, the input ingredient, and the predicted `tool`.

To isolate the challenge of planning knowledge correction, we assume a competent controller gathers all ingredients and tools; thus, an agent starts each episode with all necessary ingredients and tools. An episode (max 50 timesteps) is successful if the agent completes the plan.

### A.2  RESULTS

Table 6 shows that XENON achieves a perfect success rate ($1.00 \pm 0.00$), significantly outperforming both SC ($0.75 \pm 0.04$) and DECKARD ($0.09 \pm 0.02$). These results demonstrate that XENON's core mechanisms (ADG and FAM) are generalizable, effectively correcting flawed planning knowledge in a domain that requires the agent to discover valid symbolic actions and preconditions. Notably, the SC baseline fails to achieve high performance, even in the TextWorld Cooking environment which is simpler than Minecraft. This reinforces our claim that relying on LLM self-correction is less reliable than XENON's experience-based algorithmic knowledge correction.

## B PROMPTS AND QUALITATIVE RESULTS OF LLM SELF-CORRECTION

### B.1 DEPENDENCY CORRECTION

Figure 9 shows the prompt used for dependency correction.

```
1  You are a professional game analyst. For a given <item_name>, you need to
       make <required_items> to get the item.
2  If you make <required_items> well, I will give you 1 $.
3
4  I will give you recent transitions.
5  % Recent failed trajectories are given
6  [Failed example]
7  <item_name>: {item_name}
8  <hypothesized_required_items>: {original_prediction}
9  <inventory>: {inventory}
10 <plan>: {failed_subgoal}
11 <success>: false
12
13 I will give you learned items similar to <item_name>, and their validated
       required items, just for reference.
14 % K similar experienced items and their requirements are given
15 [Success Example]
16 <item_name>: {experienced_item}
17 <required_items> {experienced_requirements}
18
19 % Make a new predicted requirement set
20 [Your turn]
21 Here is <item_name>, you MUST output <required_items> to obtain the item
       in JSON format. Remember <required_items> MUST be in JSON format.
22
23 <item_name>: {item_name}
24 <required_items>:
```

Figure 9: Prompt used for LLM self-correction about dependencies.

We provide some examples of actual prompts and LLM outputs in Figure 10, Figure 11

### B.2 ACTION CORRECTION

Figure 12 shows the prompt used self-reflection for failed actions.

We provide some examples of actual prompts and LLM outputs in Figure 13, Figure 14

```
1   You are a professional game analyst. For a given <item_name>, you need to make <required_items
        > to get the item.
2   If you make <required_items> well, I will give you 1 $.
3
4   I will give you recent transitions.
5
6   [Failed example]
7   <item_name>: iron_nugget
8   <hypothesized_required_items>: {'iron_ore': 1, 'crafting_table': 1}
9   <inventory>: {'crafting_table': 1, 'wooden_sword': 1, 'wooden_pickaxe': 1, 'torch': 4, '
        furnace': 1, 'stone_pickaxe': 1, 'iron_axe': 1, 'iron_shovel': 1, 'stick': 2, 'iron_
        pickaxe': 1, 'diamond': 3, 'iron_ingot': 2, 'iron_ore': 2, 'gold_ore': 1, 'coal': 1}
10  <plan>: dig down and mine iron_nugget
11  <success>: false
12
13  I will give you learned items similar to <item_name>, and their validated required items, just
         for reference.
14  [Success Example]
15  <item_name>:
16  iron_ingot
17  <required_items>:
18  {'recipe': {'furnace': 1, 'iron_ore': 1, 'coals': 1}}
19  [Success Example]
20  <item_name>:
21  iron_pickaxe
22  <required_items>:
23  {'recipe': {'stick': 2, 'iron_ingot': 3, 'crafting_table': 1}}
24  [Success Example]
25  <item_name>:
26  iron_shovel
27  <required_items>:
28  {'recipe': {'stick': 2, 'iron_ingot': 1, 'crafting_table': 1}}
29
30  [Your turn]
31  Here is <item_name>, you MUST output <required_items> to obtain the item in JSON format.
        Remember <required_items> MUST be in JSON format.
32
33  <item_name>:
34  iron_nugget
35  <required_items>:
36  % LLM output: {'recipe': {'iron_ore': 1, 'crafting_table': 1}}
```

Figure 10: Example of dependency self-correction for iron_nugget.

```
1   You are a professional game analyst. For a given <item_name>, you need to make <required_items
        > to get the item.
2   If you make <required_items> well, I will give you 1 $.
3
4   I will give you recent transitions.
5
6   [Failed example]
7   <item_name>: charcoal
8   <hypothesized_required_items>: {'oak_log': 8}
9   <inventory>: {'dirt': 1, 'oak_log': 2, 'crafting_table': 1, 'wooden_hoe': 1, 'wooden_pickaxe':
        1, 'torch': 4, 'stone_axe': 1, 'furnace': 1, 'stone_pickaxe': 1, 'stick': 2, 'iron_
        pickaxe': 1, 'diamond': 1, 'iron_ingot': 3, 'iron_ore': 2, 'coal': 2}
10  <action>: craft charcoal
11  <success>: false
12
13  I will give you learned items similar to <item_name>, and their validated required items, just
        for reference.
14  [Success Example]
15  <item_name>:
16  coals
17  <required_items>:
18  {'recipe': {'wooden_pickaxe': 1}}
19  [Success Example]
20  <item_name>:
21  furnace
22  <required_items>:
23  {'recipe': {'cobblestone': 8, 'crafting_table': 1}}
24  [Success Example]
25  <item_name>:
26  diamond
27  <required_items>:
28  {'recipe': {'iron_pickaxe': 1}}
29
30  [Your turn]
31  Here is <item_name>, you MUST output <required_items> to achieve charcoal in JSON format.
        Remember <required_items> MUST be in JSON format.
32
33  <item_name>:
34  charcoal
35  <required_items>:
36  % LLM output: {'recipe': {'oak_log': 8}}
```

Figure 11: Example of dependency self-correction for charcoal.

```
1  % LLM self-reflection to analyze failure reasons
2  You are a professional game analyst.
3  For a given <item_name> and <inventory>, you need to analyze why <plan>
       failed to get the item.
4  I will give you examples of analysis as follow.
5
6  [Example]
7  <item_name>: wooden_pickaxe
8  <inventory>: {'stick': 4, 'planks': 4, 'crafting_table': 1}
9  <plan>: smelt wooden_pickaxe
10 <failure_analysis>
11 {"analysis": "You failed because you cannot smelt a wooden_pickaxe. You
       should craft it instead."}
12
13 [Example]
14 <item_name>: stone_pickaxe
15 <inventory>: {'stick': 4, 'planks': 4, 'crafting_table': 1}
16 <plan>: craft stone_pickaxe
17 <failure_analysis>
18 {"analysis": "You failed because you do not have enough cobblestones."}
19
20 [Your turn]
21 Here is <item_name>, <inventory> and <plan>, you MUST output <failure_
       analysis> concisely in JSON format.
22
23 <item_name>: {item_name}
24 <inventory>: {inventory}
25 <plan>: {plan}
26 <failure_analysis>
27
28 % Then, using the self-reflection results, LLM self-correct its actions.
29 For an item name, you need to make a plan, by selecting one among
       provided options.
30 I will give you examples of which plans are needed to achieve an item,
       just for reference.
31 [Example]
32 <item name>
33 {similar_item}
34 <task planning>
35 {successful_plan
36
37 Here are some analyses on previous failed plans for this item.
38 [Analysis]
39 {'item_name': {item}, 'inventory': {inventory}, 'plan': '{plan}', '
       failure_analysis': '{self-reflection}'}
40
41 [Your turn]
42 Here is <item name>, you MUST select one from below <options>, to make <
       task planning>.
43 you MUST select one from below <options>. DO NOT MAKE A PLAN NOT IN <
       options>.
44
45 <options>:
46 1: {"task": "dig down and mine {item}", "goal": [{item}, {quantity}]]}
47 2: {"task": "craft {item}", "goal": [{item}, {quantity}]]}
48 3: {"task": "smelt {item}", "item": [{item}, {quantity}]}
49
50 <item name>
51 {item}
52 <task planning>
```

Figure 12: Prompts used for LLM self-correction about actions.

```
1    For an item name, you need to make a plan, by selecting one among provided options.
2    I will give you examples of which plans are needed to achieve an item, just for reference.
3
4    [Example]
5    <item name>
6    iron_ingot
7    <task planning>
8    {"task": "smelt iron_ingot", "goal": ["iron_ingot", 1]}
9
10   [Example]
11   <item name>
12   iron_pickaxe
13   <task planning>
14   {"task": "craft iron_pickaxe", "goal": ["iron_pickaxe", 1]}
15
16   [Example]
17   <item name>
18   iron_shovel
19   <task planning>
20   {"task": "craft iron_shovel", "goal": ["iron_shovel", 1]}
21
22   Here are some analyses on previous failed plans for this item.
23   [Analysis]
24   {'item_name': 'iron_nugget',
25   'inventory': {'crafting_table': 1, 'wooden_sword': 1, 'wooden_pickaxe': 1, 'torch': 4, '
         furnace': 1, 'stone_pickaxe': 1, 'iron_axe': 1, 'iron_shovel': 1, 'stick': 2, 'iron_
         pickaxe': 1, 'diamond': 3, 'iron_ingot': 2, 'iron_ore': 2, 'gold_ore': 1, 'coal': 1},
26   'plan': 'dig down and mine iron_nugget',
27   'failure_analysis': 'You failed because you do not have any iron ore or diamond ore to mine
         for iron nuggets.'}
28
29   [Your turn]
30   Here is <item name>, you MUST select one from below <options>, to make <task planning>.
31   you MUST select one from below <options>. DO NOT MAKE A PLAN NOT IN <options>.
32
33   <options>
34   1. {"task": "dig down and mine iron_nugget", "goal": ["iron_nugget", 1]}
35   2. {"task": "craft iron_nugget", "goal": ["iron_nugget", 1]}
36   3. {"task": "smelt iron_nugget", "goal": ["iron_nugget", 1]}
37
38   <item name>
39   iron_nugget
40   % LLM output: '{"task": "dig down and mine iron_nugget", "goal": ["iron_nugget", 1]}'
```

Figure 13: Example of action self-correction for iron_nugget.

```
1    For an item name, you need to make a plan, by selecting one among provided options.
2    I will give you examples of which plans are needed to achieve an item, just for reference.
3
4    [Example]
5    <item name>
6    coals
7    <task planning>
8    {"task": "dig down and mine coals", "goal": ["coals", 1]}
9
10   [Example]
11   <item name>
12   furnace
13   <task planning>
14   {"task": "craft furnace", "goal": ["furnace", 1]}
15
16   [Example]
17   <item name>
18   diamond
19   <task planning>
20   {"task": "dig down and mine diamond", "goal": ["diamond", 1]}
21
22   Here are some analyses on previous failed plans for this item.
23   [Analysis]
24   {'item_name': 'charcoal',
25   'inventory': {'dirt': 1, 'oak_log': 2, 'crafting_table': 1, 'wooden_hoe': 1, 'wooden_pickaxe':
            1, 'torch': 4, 'stone_axe': 1, 'furnace': 1, 'stone_pickaxe': 1, 'stick': 2, 'iron_
            pickaxe': 1, 'diamond': 1, 'iron_ingot': 3, 'iron_ore': 2, 'coal': 2},
26   'plan': 'mine iron_nugget',
27   'failure_analysis': 'You failed because you already have enough charcoal.'}
28
29
30   [Your turn]
31   Here is <item name>, you MUST select one from below <options>, to make <task planning>.
32   you MUST select one from below <options>. DO NOT MAKE A PLAN NOT IN <options>.
33
34   <options>
35   1. {"task": "mine iron_nugget", "goal": ["charcoal", 1]}
36   2. {"task": "craft charcoal", "goal": ["charcoal", 1]}
37   3. {"task": "smelt charcoal", "goal": ["charcoal", 1]}
38
39   <item name>
40   charcoal
41   <task planning>
42   % LLM output: '{"task": "craft charcoal", "goal": ["charcoal", 1]}'
```

Figure 14: Example of action self-correction for charcoal.

## C  EXPERIENCED REQUIREMENT SET AND DEPENDENCY GRAPH UPDATE

We note that the assumptions explained in this section are largely similar to those in the implementation of DECKARD (Nottingham et al., 2023)[2].

**Determining experienced requirement set.** When the agent obtains item $v$ while executing a subgoal $(a, q, u)$, it determines the experienced requirement set $\mathcal{R}_{exp}(v)$ differently depending on whether the high-level action $a$ is "mine" or falls under "craft" or "smelt". If $a$ is "mine", the agent determines $\mathcal{R}_{exp}(v)$ based on the pickaxe in its inventory. If no pickaxe is held, $\mathcal{R}_{exp}(v)$ is $\emptyset$. Otherwise, $\mathcal{R}_{exp}(v)$ becomes $\{(\text{the highest-tier pickaxe the agent has}, 1)\}$, where the highest-tier pickaxe is determined following the hierarchy: "wooden_pickaxe", "stone_pickaxe", "iron_pickaxe", "diamond_pickaxe". If $a$ is "craft" or "smelt", the agent determines the used items and their quantities as $\mathcal{R}_{exp}(v)$ by observing inventory changes when crafting or smelting $v$.

**Dependency graph update.** When the agent obtains an item $v$ and its $\mathcal{R}_{exp}(v)$ for the first time, it updates its dependency graph $\hat{\mathcal{G}} = (\hat{\mathcal{V}}, \hat{\mathcal{E}})$. Since $\mathcal{R}_{\exp}(v)$ only contains items acquired before $v$, no cycles can be introduced to ADG during learning. The update proceeds as follows: The agent adds $v$ to both the set of known items $\hat{\mathcal{V}}$. Then, it updates the edge set $\hat{\mathcal{E}}$ by replacing $v$'s incoming edges with $\mathcal{R}_{exp}(v)$: it removes all of $v$'s incoming edges $(u, \cdot, v) \in \hat{\mathcal{E}}$ and adds new edges $(u_i, q_i, v)$ to $\hat{\mathcal{E}}$ for every $(u_i, q_i) \in \mathcal{R}_{exp}(v)$.

---

[2]https://github.com/DeckardAgent/deckard

---

**Algorithm 1:** Pseudocode of XENON

---

**input :** invalid action threshold $x_0$, inadmissible item threshold $c_0$, less-explored item scale $\alpha_s$, inadmissible item scale $\alpha_i$

1 Initialize dependency $\hat{\mathcal{G}} \leftarrow (\hat{\mathcal{V}}, \hat{\mathcal{E}})$, revision counts $C[v] \leftarrow 1$ for all $v \in \hat{\mathcal{V}}$

2 Initialize memory $S(a,v) = 0, F(a,v) = 0$ for all $v \in \hat{\mathcal{V}}, a \in \mathcal{A}$

3 **while** *learning* **do**

4     Get an empty inventory $inv$

5     $v_g \leftarrow$ `SelectGoalWithDifficulty`$(\hat{\mathcal{G}}, C[\cdot])$          `// DEX Appendix G`

6     **while** $H_{episode}$ **do**

7         **if** $v_g \in inv$ **then**

8             $v_g \leftarrow$ `SelectGoalWithDifficulty`$(\hat{\mathcal{G}}, C[\cdot])$

9         Series of aggregated requirements $((q_l, u_l))_{l=1}^{L_{v_g}}$ using $\hat{\mathcal{G}}$ and $inv$
        `// from Section 3`

10         Plan $P \leftarrow ((a_l, q_l, u_l))_{l=1}^{L_{v_g}}$ by selecting $a_l$ for each $u_l$, using LLM, $S$, $F$, $x_0$

11         **foreach** *subgoal* $(a, q, u) \in P$ **do**

12             Execute $(a, q, u)$ then get the execution result *success*

13             Get an updated inventory $inv$, dependency graph $\hat{\mathcal{G}}$     `// from Section 3`

14             **if** *success* **then** $S(a, u) \leftarrow S(a, u) + 1$

15             **else** $F(a, u) \leftarrow F(a, u) + 1$

16             **if** *not success* **then**

17                 **if** `All actions are invalid` **then**

18                     $\hat{\mathcal{G}}, C \leftarrow$ `RevisionByAnalogy`$(\hat{\mathcal{G}}, u, C[\cdot], c_0, \alpha_s, \alpha_i)$
                    `// ADG Section 4.1`

19                     Reset memory $S(\cdot, u) \leftarrow 0, F(\cdot, u) \leftarrow 0$

20                     $v_g \leftarrow$ `SelectGoalWithDifficulty`$(\hat{\mathcal{G}}, C[\cdot])$

21                 **break**

---

# D   Full procedure of XENON

The full procedure of XENON is outlined in Algorithm 1

# E  DETAILS IN ADAPTIVE DEPENDENCY GRAPH (ADG)

## E.1  RATIONALE FOR INITIAL KNOWLEDGE

In real-world applications, a human user may wish for an autonomous agent to accomplish certain goals, yet the user themselves may have limited or no knowledge of how to achieve them within a complex environment. We model this scenario by having a user specify goal items without providing the detailed requirements, and then the agent should autonomously learn how to obtain these goal items. The set of 67 goal item names ($\mathcal{V}_0$) provided to the agent represents such user-specified goal items, defining the learning objectives.

To bootstrap learning in complex environments, LLM-based planning literature often utilizes minimal human-written plans for initial knowledge (Zhao et al., 2024; Chen et al., 2024). In our case, we provide the agent with 3 human-written plans (shown in Appendix I). By executing these plans, our agent can experience items and their dependencies, thereby bootstrapping the dependency learning process.

## E.2  DETAILS IN DEPENDENCY GRAPH INITIALIZATION

**Keeping ADG acyclic during initialization.** During initialization, XENON prevents cycles in ADG algorithmically and maintains ADG as a directed acyclic graph, by, whenever adding an LLM-predicted requirement set for an item, discarding any set that would make a cycle and instead assign an empty requirement set to that item. Specifically, we identify and prevent cycles in three steps when adding LLM-predicted incoming edges for an item $v$. First, we tentatively insert the LLM-predicted incoming edges of $v$ into the current ADG. Second, we detect cycles by checking whether any of $v$'s parents now appears among $v$'s descendants in the updated graph. Third, if a cycle is detected, we discard the LLM-predicted incoming edges for $v$ and instead assign an empty set of incoming edges to $v$ in the ADG.

Pseudocode is shown in Algorithm 2. The prompt is shown in Figure 15.

---

**Algorithm 2:** GraphInitialization

---

**input**   : Goal items $\mathcal{V}_0$, (optional) human written plans $\mathcal{P}_0$

**output** : Initialized dependency graph $\hat{\mathcal{G}} = (\hat{\mathcal{V}}, \hat{\mathcal{E}})$, experienced items $\mathcal{V}$

1  Initialize a set of known items $\hat{\mathcal{V}} \leftarrow \mathcal{V}_0$, edge set $\hat{\mathcal{E}} \leftarrow \emptyset$

2  Initialize a set of experienced items $\mathcal{V} \leftarrow \emptyset$

3  **foreach** *plan in $\mathcal{P}_0$* **do**

4   Execute the plan and get experienced items and their experienced requirement sets
  $\left\{ (v_n, \mathcal{R}_{exp}(v_n)) \right\}_{n=1}^{N}$

5   **foreach** $(v, \mathcal{R}_{exp}(v)) \in \left\{ (v_n, \mathcal{R}_{exp}(v_n)) \right\}_{n=1}^{N}$ **do**

6    **if** $v \notin \mathcal{V}$ **then**
    /* graph update from Appendix C                              */

7     $\mathcal{V} \leftarrow \mathcal{V} \cup \{v\}, \hat{\mathcal{V}} \leftarrow \hat{\mathcal{V}} \cup \{v\}$

8     Add edges to $\hat{\mathcal{E}}$ according to $\mathcal{R}_{exp}(v)$

 /* Graph construction using LLM predictions                       */

9  **while** $\exists v \in \hat{\mathcal{V}} \setminus \mathcal{V}$ *whose requirement set $\mathcal{R}(v)$ has not yet been predicted by the LLM* **do**

10   Select such an item $v \in \hat{\mathcal{V}} \setminus \mathcal{V}$ (i.e., $\mathcal{R}(v)$ has not yet been predicted)

11   Select $\mathcal{V}_K \subseteq \mathcal{V}$ based on Top-K semantic similarity to $v$, $|\mathcal{V}_K| = K$

12   Predict $\mathcal{R}(v) \leftarrow LLM(v, \{(u, \mathcal{R}(u, \hat{\mathcal{G}}))\}_{u \in \mathcal{V}_K})$

13   **foreach** $(u_j, q_j) \in \mathcal{R}(v)$ **do**

14    $\hat{\mathcal{E}} \leftarrow \hat{\mathcal{E}} \cup \{(u_j, q_j, v)\}$

15    **if** $u_j \notin \hat{\mathcal{V}}$ **then**

16     $\hat{\mathcal{V}} \leftarrow \hat{\mathcal{V}} \cup \{u_j\}$

---

```
1  You are a professional game analyst. For a given <item_name>, you need to
       make <required_items> to get the item.
2  If you make <required_items> well, I will give you 1 $.
3
4  I will give you some examples <item_name> and <required_items>.
5
6  [Example] % TopK similar experienced items are given as examples
7  <item_name>: {experienced_item}
8  <required_items>: {experienced_requirement_set}
9
10 [Your turn]
11 Here is a item name, you MUST output <required_items> in JSON format.
       Remember <required_items> MUST be in JSON format.
12
13 <item_name>: {item_name}
14 <required_items>:
```

Figure 15: Prompt for requirement set prediction for dependency graph initialization

### E.3 PSEUDOCODE OF REVISIONBYANALOGY

Pseudocode is shown in Algorithm 3.

---

**Algorithm 3:** RevisionByAnalogy

---

**input** : Dependency graph $\hat{\mathcal{G}} = (\hat{\mathcal{V}}, \hat{\mathcal{E}})$, an item to revise $v$, exploration counts $C[\cdot]$, inadmissible item threshold $c_0$, less-explored item scale $\alpha_s$, inadmissible item scale $\alpha_i$

**output** : Revised dependency graph $\hat{\mathcal{G}} = (\hat{\mathcal{V}}, \hat{\mathcal{E}})$, exploration counts $C[\cdot]$

1 Consider cases based on $C[v]$:

2 **if** $C[v] > c_0$ **then**

```
/* v is inadmissible                                        */
/* resource set:  items previously consumed for crafting
   other items                                              */
```

3     $\mathcal{R}(v) \leftarrow \{(u, \alpha_i) \mid u \in \text{"resource" set}\}$

```
/* Remove all incoming edges to v in Ê and add new edges    */
```

4     $\hat{\mathcal{E}} \leftarrow \hat{\mathcal{E}} \setminus \{(x, q, v) \mid (x, q, v) \in \hat{\mathcal{E}}\}$

5     **foreach** $(u, \alpha_i) \in \mathcal{R}(v)$ **do**

6         $\hat{\mathcal{E}} \leftarrow \hat{\mathcal{E}} \cup \{(u, \alpha_i, v)\}$

```
/* Revise requirement sets of descendants of v             */
```

7     Find the set of all descendants of $v$ in $\hat{\mathcal{G}}$ (excluding $v$): $\mathcal{W} \leftarrow \text{FindAllDescendants}(v, \hat{\mathcal{G}})$

8     **for** *each item $w$ in $\mathcal{W}$* **do**

9         Invoke RevisionByAnalogy for $w$

10 **else**

```
/* v is less explored yet.  Revise based on analogy        */
```

11     Find similar successfully obtained items $\mathcal{V}_K \subseteq \hat{\mathcal{V}}$ based on Top-K semantic similarity to $v$

12     Candidate items $U_{cand} \leftarrow \{u \mid \exists w \in \mathcal{V}_K, (u, \cdot, w) \in \hat{\mathcal{E}}\}$ /* all items required to obtain similar successfully obtained items $\mathcal{V}_K$ */

13     Start to construct a requirement set, $\mathcal{R}(v) \leftarrow \emptyset$

14     **for** *each item $u$ in $U_{cand}$* **do**

15         **if** *$u$ is in "resource" set* **then**

16             Add $(u, \alpha_s \times C[v])$ to $\mathcal{R}(v)$

17         **else**

18             Add $(u, 1)$ to $\mathcal{R}(v)$

19     Update $\hat{\mathcal{G}}$: Remove all incoming edges to $v$ in $\hat{\mathcal{E}}$, and add new edges $(u, q, v)$ to $\hat{\mathcal{E}}$ for each $(u, q) \in \mathcal{R}(v)$

---

# F   STEP-BY-STEP PLANNING USING FAM

Given a sequence of aggregated requirements $((q_l, v_l))_{l=1}^{L}$, XENON employs a step-by-step planning approach, iteratively selecting an high-level action $a_l$ for each requirement item $v_l$ to make a subgoal $(a_l, q_l, v_l)$. This process considers the past attempts to obtain $v_l$ using specific actions. Specifically, for a given item $v_l$, if FAM has an empirically valid action, XENON reuses it without prompting the LLM. Otherwise, XENON prompts the LLM to select an action, leveraging information from (i) valid actions for items semantically similar to $v_l$, (ii) empirically invalid actions for $v_l$.

The pseudocode for this action selection process is detailed in Algorithm 4. The prompt is shown in Figure 16.

---

**Algorithm 4:** Step-by-step Planning with FAM

**Input**   : An item $v$, Action set $\mathcal{A}$, Success/Failure counts from FAM $S(\cdot, \cdot)$ and $F(\cdot, \cdot)$, Invalid action threshold $x_0$

**Output** : Selected action $a_{selected}$

```
/* 1.  Classify actions based on FAM history (S and F counts)
   */
```
1   $\mathcal{A}_v^{valid} \leftarrow \{a \in \mathcal{A} \mid S(a, v) > 0 \land S(a, v) > F(a, v) - x_0\}$
2   $\mathcal{A}_v^{invalid} \leftarrow \{a \in \mathcal{A} \mid F(a, v) \geq S(a, v) + x_0\}$
3   **if** $\mathcal{A}_v^{valid} \neq \emptyset$ **then**
```
    /* Reuse the empirically valid action if it exists      */
```
4   $\quad$ Select $a_{selected}$ from $\mathcal{A}_v^{valid}$
5   $\quad$ **return** $a_{selected}$
6   **else**
```
    /* Otherwise, query LLM with similar examples and filtered
       candidates                                            */
    /* (i) Retrieve valid actions from other items for examples
       */
```
7   $\quad$ $\mathcal{V}_{source} \leftarrow \{u \in \hat{V} \setminus \{v\} \mid \exists a', S(a', u) > 0 \land S(a', u) > F(a', u) - x_0\}$
8   $\quad$ Identify $\mathcal{V}_{topK} \subseteq \mathcal{V}_{source}$ as the $K$ items most similar to $v$ (using S-BERT)
9   $\quad$ $\mathcal{D}_{examples} \leftarrow \{(u, a_{valid}) \mid u \in \mathcal{V}_{topK}, a_{valid} \in \mathcal{A}_u^{valid}\}$
```
    /* (ii) Prune invalid actions to form candidates          */
```
10  $\quad$ $\mathcal{A}_v^{cand} \leftarrow \mathcal{A} \setminus \mathcal{A}_v^{invalid}$
11  $\quad$ **if** $\mathcal{A}_v^{cand} = \emptyset$ **then**
12  $\quad\quad$ $\mathcal{A}_v^{cand} \leftarrow \mathcal{A}$
13  $\quad$ $a_{selected} \leftarrow \text{LLM}(v, \mathcal{D}_{examples}, \mathcal{A}_v^{cand})$
14  $\quad$ **return** $a_{selected}$

---

```
1   For an item name, you need to make a plan, by selecting one among
        provided options.
2   I will give you examples of which plans are needed to achieve an item,
        just for reference.
3
4   % Similar items and their successful plans are given
5   [Example]
6   <item name>
7   {similar_item}
8   <task planning>
9   {successful_plan}
10
11  [Your turn]
12  Here is <item name>, you MUST select one from below <options>, to make <
        task planning>.
13  you MUST select one from below <options>. DO NOT MAKE A PLAN NOT IN <
        options>.
14
15  % Three actions are given, excluding any that were empirically invalid
16  <options>:
17  1: {"task": "dig down and mine {item}", "goal": [{item}, {quantity}]]}
18  2: {"task": "craft {item}", "goal": [{item}, {quantity}]]}
19  3: {"task": "smelt {item}", "item": [{item}, {quantity}]}
20
21  <item name>
22  {item}
23  <task planning>
```

Figure 16: Prompt for action selection

## G  DIFFICULTY-BASED EXPLORATION (DEX)

For autonomous dependency learning, we introduce DEX. DEX strategically selects items that (1) appear easier to obtain, prioritizing those (2) under-explored for diversity and (3) having fewer immediate prerequisite items according to the learned graph $\hat{\mathcal{G}}$. (line 5 in Algorithm 1). First, DEX defines the previously unobtained items but whose required items are all obtained according to learned dependency $\hat{\mathcal{G}}$ as the frontier $F$. Next, the least explored frontier set $\mathcal{F}_{min} \coloneqq \{f \in F \mid C(f) = \min_{f' \in F} C(f')\}$ is identified, based on revision counts $C(\cdot)$. For items $f' \in \mathcal{F}_{min}$, difficulty $D(f')$ is estimated as $L_{f'}$, the number of distinct required items needed to obtain $f'$ according to $\hat{\mathcal{G}}$. The intrinsic goal $g$ is then selected as the item in $\mathcal{F}_{min}$ with the minimum estimated difficulty: $g = \arg\min_{f' \in \mathcal{F}_{min}} D(f')$. Ties are broken uniformly at random.

While our frontier concept is motivated by DECKARD (Nottingham et al., 2023), DEX's selection process differs significantly. DECKARD selects randomly from $\{v \in \mathcal{F} \mid C(v) \le c_0\}$, but if this set is empty, it selects randomly from the union of frontier set and previously obtained item set. This risks inefficient attempts on already obtained items. In contrast, DEX exclusively selects goals from $\mathcal{F}_{min}$, inherently avoiding obtained items. This efficiently guides exploration towards achievable, novel dependencies.

## H    CONTEXT-AWARE REPROMPTING (CRE)

Minecraft, a real-world-like environment can lead to situations where the controller stalls (e.g., when stuck in deep water or a cave). To assist the controller, the agent provides temporary prompts to guide it (e.g., "get out of the water and find trees"). XENON proposes a context-aware reprompting scheme. It is inspired by Optimus-1 Li et al. (2024b) but introduces two key differences:

(a) **Two-stage reasoning.**   When invoked, in Optimus-1, LLM simultaneously interprets image observations, decides whether to reprompt, and generates new prompts. XENON decomposes this process into two distinct steps:

   (i) the LLM generates a caption for the current image observation, and
   (ii) using *text-only* input (the generated caption and the current subgoal prompt), the LLM determines if reprompting is necessary and, if so, produces a temporary prompt.

(b) **Trigger.**  Unlike Optimus-1, which invokes the LLM at fixed intervals, XENON calls the LLM only if the current subgoal item has not been obtained within that interval.  This approach avoids unnecessary or spurious interventions from a smaller LLM.

The prompt is shown in Figure 17.

```
1   % Prompt for the first step: image captioning
2   Given a Minecraft game image, describe nearby Minecraft objects, like
        tree, grass, cobblestone, etc.
3   [Example]
4   "There is a large tree with dark green leaves surrounding the area."
5   "The image shows a dark, cave-like environment in Minecraft. The player
        is digging downwards. There are no visible trees or grass in this
        particular view."
6   "The image shows a dark, narrow tunnel made of stone blocks. The player
        is digging downwards."
7   [Your turn]
8   Describe the given image, simply and clearly like the examples.
9
10  % Prompt for the second step: reasoning whether reprompting is needed or
        not
11  Given <task> and <visual_description>, determine if the player needs
        intervention to achieve the goal. If intervention is needed, suggest
        a task that the player should perform.
12  I will give you examples.
13  [Example]
14  <task>: chop tree
15  <visual_description>: There is a large tree with dark green leaves
        surrounding the area.
16  <goal_item>: logs
17  <reasoning>:
18  {{
19      "need_intervention": false,
20      "thoughts": "The player can see a tree and can chop it down to get
            logs.",
21      "task": "",
22  }}
23  [Example]
24  <task>: chop tree
25  <visual_description>: The image shows a dirt block in Minecraft. There is
         a tree in the image, but it is too far from here.
26  <goal_item>: logs
27  <reasoning>:
28  {{
29      "need_intervention": true,
30      "thoughts": "The player is far from trees. The player needs to move
            to the trees.",
31      "task": "explore to find trees",
32  }}
33  [Example]
34  <task>: dig down to mine iron_ore
35  <visual_description>: The image shows a dark, narrow tunnel made of stone
         blocks. The player is digging downwards.
36  <goal_item>: iron_ore
37  <reasoning>:
38  {{
39      "need_intervention": false,
40      "thoughts": "The player is already digging down and is likely to find
             iron ore.",
41      "task": "",
42  }}
43  [Your turn]
44  Here is the <task>, <visual_description>, and <goal_item>.
45  You MUST output the <reasoning> in JSON format.
46  <task>: {task} % current prompt for the controller
47  <visual_description>: {visual_description} % caption from the step 1
48  <goal_item>: {goal_item} % current subgoal item
49  <reasoning>:
```

Figure 17: Prompt for context-aware reprompting

## I  IMPLEMENTATION DETAILS

To identify similar items, semantic similarity between two items is computed as the cosine similarity of their Sentence-BERT (all-MiniLM-L6-v2 model) embeddings (Reimers & Gurevych, 2019). This metric is utilized whenever item similarity comparisons are needed, such as in Algorithm 2, Algorithm 3, and Algorithm 4.

### I.1  HYPERPARAMETERS

Table 7: Hyperparameters used in our experiments.

| Hyperparameter | Notation | Value |
|---|---|---|
| Failure threshold for invalid action | $x_0$ | 2 |
| Revision count threshold for inadmissible items | $c_0$ | 3 |
| Required items quantity scale for less explored items | $\alpha_s$ | 2 |
| Required items quantity scale for inadmissible items | $\alpha_i$ | 8 |
| Number of top-K similar experienced items used | $K$ | 3 |

For all experiments, we use consistent hyperparameters across environments. The hyperparameters, whose values are determined with mainly considering robustness against imperfect controllers. All hyperparameters are listed in Table 7. The implications of increasing each hyperparameter's value are detailed below:

- $x_0$ (failure threshold for empirically invalid action): Prevents valid actions from being misclassified as invalid due to accidental failures from an imperfect controller or environmental stochasticity. Values that are too small or large hinder dependency learning and planning by hampering the discovery of valid actions.

- $c_0$ (exploration count threshold for inadmissible items): Ensures an item is sufficiently attempted before being deemed 'inadmissible' and triggering a revision for its descendants. Too small/large values could cause inefficiency; small values prematurely abandon potentially correct LLM predictions for descendants, while large values prevent attempts on descendant items.

- $\alpha_s$ (required items quantity scale for less explored items): Controls the gradual increase of required quantities for revised required items. Small values make learning inefficient by hindering item obtaining due to insufficient required items, yet large values lower robustness by overburdening controllers with excessive quantity demands.

- $\alpha_i$ (required items quantity scale for inadmissible items): Ensures sufficient acquisition of potential required items before retrying inadmissible items to increase the chance of success. Improper values reduce robustness; too small leads to failure in obtaining items necessitating many items; too large burdens controllers with excessive quantity demands.

- $K$ (Number of similar items to retrieve): Determines how many similar, previously successful experiences are retrieved to inform dependency revision (Algorithm 3) and action selection (Algorithm 4).

### I.2  HUMAN-WRITTEN PLANS

We utilize three human-written plans (for iron sword, golden sword, and diamond, shown in Plan 18, 19, and 20, respectively), the format of which is borrowed from the human-written plan examples in the publicly released Optimus-1 repository [3]. We leverage the experiences gained from executing these plans to initialize XENON's knowledge.

---

[3]`https://github.com/JiuTian-VL/Optimus-1/blob/main/src/optimus1/example.py`

```
1   iron_sword: str = """
2   <goal>: craft an iron sword.
3   <requirements>:
4   1. log: need 7
5   2. planks: need 21
6   3. stick: need 5
7   4. crafting_table: need 1
8   5. wooden_pickaxe: need 1
9   6. cobblestone: need 11
10  7. furnace: need 1
11  8. stone_pickaxe: need 1
12  9. iron_ore: need 2
13  10. iron_ingot: need 2
14  11. iron_sword: need 1
15  <plan>
16  {
17  "step 1": {"prompt": "mine logs", "item": ["logs", 7]},
18  "step 2": {"prompt": "craft planks", "item": ["planks", 21]},
19  "step 3": {"prompt": "craft stick", "item": ["stick", 5]},
20  "step 4": {"prompt": "craft crafting_table", "item": ["crafting_table",
        1]},
21  "step 5": {"prompt": "craft wooden_pickaxe", "item": ["wooden_pickaxe",
        1]},
22  "step 6": {"prompt": "mine cobblestone", "item": ["cobblestone", 11]},
23  "step 7": {"prompt": "craft furnace", "item": ["furnace", 1]},
24  "step 8": {"prompt": "craft stone_pickaxe", "item": ["stone_pickaxe",
        1]},
25  "step 9": {"prompt": "mine iron_ore", "item": ["iron_ore", 2]},
26  "step 10": {"prompt": "smelt iron_ingot", "item": ["iron_ingot", 2]},
27  "step 11": {"prompt": "craft iron_sword", "item": ["iron_sword", 1]}
28  }
29  """
```

Figure 18: Human-written plan for crafting an iron sword.

```
1  golden_sword: str = """
2  <goal>: craft a golden sword.
3  <requirements>:
4  1. log: need 9
5  2. planks: need 27
6  3. stick: need 7
7  4. crafting_table: need 1
8  5. wooden_pickaxe: need 1
9  6. cobblestone: need 11
10 7. furnace: need 1
11 8. stone_pickaxe: need 1
12 9. iron_ore: need 3
13 10. iron_ingot: need 3
14 11. iron_pickaxe: need 1
15 12. gold_ore: need 2
16 13. gold_ingot: need 2
17 14. golden_sword: need 1
18 <plan>
19 {
20 "step 1": {"prompt": "mine logs", "item": ["logs", 7]},
21 "step 2": {"prompt": "craft planks", "item": ["planks", 21]},
22 "step 3": {"prompt": "craft stick", "item": ["stick", 5]},
23 "step 4": {"prompt": "craft crafting_table", "item": ["crafting_table",
      1]},
24 "step 5": {"prompt": "craft wooden_pickaxe", "item": ["wooden_pickaxe",
      1]},
25 "step 6": {"prompt": "mine cobblestone", "item": ["cobblestone", 11]},
26 "step 7": {"prompt": "craft furnace", "item": ["furnace", 1]},
27 "step 8": {"prompt": "craft stone_pickaxe", "item": ["stone_pickaxe",
      1]},
28 "step 9": {"prompt": "mine iron_ore", "item": ["iron_ore", 3]},
29 "step 10": {"prompt": "smelt iron_ingot", "item": ["iron_ingot", 3]},
30 "step 11": {"task": "craft iron_pickaxe", "goal": ["iron_pickaxe", 1]},
31 "step 12": {"prompt": "mine gold_ore", "item": ["gold_ore", 2]},
32 "step 13": {"prompt": "smelt gold_ingot", "item": ["gold_ingot", 2]},
33 "step 14": {"task": "craft golden_sword", "goal": ["golden_sword", 1]}
34 }
35 """
```

Figure 19: Human-written plan for crafting a golden sword.

```
1  diamond: str = """
2  <goal>: mine a diamond.
3  <requirements>:
4  1. log: need 7
5  2. planks: need 21
6  3. stick: need 6
7  4. crafting_table: need 1
8  5. wooden_pickaxe: need 1
9  6. cobblestone: need 11
10 7. furnace: need 1
11 8. stone_pickaxe: need 1
12 9. iron_ore: need 3
13 10. iron_ingot: need 3
14 11. iron_pickaxe: need 1
15 12. diamond: need 1
16 <plan>
17 {
18 "step 1": {"prompt": "mine logs", "item": ["logs", 7]},
19 "step 2": {"prompt": "craft planks", "item": ["planks", 21]},
20 "step 3": {"prompt": "craft stick", "item": ["stick", 5]},
21 "step 4": {"prompt": "craft crafting_table", "item": ["crafting_table",
       1]},
22 "step 5": {"prompt": "craft wooden_pickaxe", "item": ["wooden_pickaxe",
       1]},
23 "step 6": {"prompt": "mine cobblestone", "item": ["cobblestone", 11]},
24 "step 7": {"prompt": "craft furnace", "item": ["furnace", 1]},
25 "step 8": {"prompt": "craft stone_pickaxe", "item": ["stone_pickaxe",
       1]},
26 "step 9": {"prompt": "mine iron_ore", "item": ["iron_ore", 2]},
27 "step 10": {"prompt": "smelt iron_ingot", "item": ["iron_ingot", 2]},
28 "step 11": {"prompt": "craft iron_pickaxe", "item": ["iron_pickaxe", 1]},
29 "step 12": {"prompt": "mine diamond", "item": ["diamond", 1]}
30 }
31 """
```

Figure 20: Human-written plan for mining a diamond.

# J DETAILS FOR EXPERIMENTAL SETUP

## J.1 COMPARED BASELINES FOR DEPENDENCY LEARNING

We compare our proposed method, XENON, against four baselines: LLM self-correction (SC), DECKARD Nottingham et al. (2023), ADAM (Yu & Lu, 2024), and RAND (the simplest baseline). As no prior baselines were evaluated under our specific experimental setup (i.e., empty initial inventory, pre-trained low-level controller), we adapted their implementation to align with our environment. SC is implemented following common methods that prompt the LLM to correct its own knowledge upon plan failures (Shinn et al., 2023; Stechly et al., 2024). A summary of all methods compared in our experiments is provided in Table 8. All methods share the following common experimental setting: each episode starts with an initial experienced requirements for some items, derived from human-written plans (details in Appendix I). Additionally, all agents begin each episode with an initial empty inventory.

Table 8: Summary of methods compared in our experiments.

| Method | Predicted Requirement Set | Action Memory | Intrinsic Goal Selection |
|---|---|---|---|
| XENON | LLM-generated (with revision) | Success & Failure | DEX |
| SC | LLM-generated (with revision) | Success & Failure | Random |
| ADAM Yu & Lu (2024) | "8 × resources" | Success-only | Random |
| DECKARD Nottingham et al. (2023) | LLM-generated (without revision) | Success-only | Frontier + obtained items |
| RAND | LLM-generated (without revision) | None | Random |

**LLM self-correction (SC).** While no prior work specifically uses LLM self-correction to learn Minecraft item dependencies in our setting, we include this baseline to demonstrate the unreliability of this approach. For predicted requirements, similar to XENON, SC initializes its dependency graph with LLM-predicted requirements for each item. When a plan for an item fails repeatedly, it attempts to revise the requirements using LLM. SC prompts the LLM itself to perform the correction, providing it with recent trajectories and the validated requirements of similar, previously obtained items in the input prompt. SC's action memory stores both successful and failed actions for each item. Upon a plan failure, the LLM is prompted to self-reflect on the recent trajectory to determine the cause of failure. When the agent later plans to obtain an item on which it previously failed, this reflection is included in the LLM's prompt to guide its action selection. Intrinsic goals are selected randomly from the set of previously unobtained items. The specific prompts used for the LLM self-correction and self-reflection in this baseline are provided in Appendix B.

**DECKARD.** The original DECKARD utilizes LLM-predicted requirements for each item but does not revise these initial predictions. It has no explicit action memory for the planner; instead, it trains and maintains specialized policies for each obtained item. It selects an intrinsic goal randomly from less explored frontier items (i.e., $\{v \in \mathcal{F} \mid C(v) \le c_0\}$). If no such items are available, it selects randomly from the union of experienced items and all frontier items.

In our experiments, the DECKARD baseline is implemented to largely mirror the original version, with the exception of its memory system. Its memory is implemented to store only successful actions without recording failures. This design choice aligns with the original DECKARD's approach, which, by only learning policies for successfully obtained items, lacks policies for unobtained items.

**ADAM.** The original ADAM started with an initial inventory containing 32 quantities of experienced resource items (i.e., items used for crafting other items) and 1 quantity of tool items (e.g., pickaxes, crafting table), implicitly treating those items as a predicted requirement set for each item. Its memory recorded which actions were used for each subgoal item without noting success or failure, and its intrinsic goal selection was guided by an expert-defined exploration curriculum.

In our experiments, ADAM starts with an empty initial inventory. The predicted requirements for each goal item in our ADAM implementation assume a fixed quantity of 8 for all resource items. This quantity was chosen to align with $\alpha_i$, the hyperparameter for the quantity scale of requirement items for inadmissible items, thereby ensuring a fair comparison with XENON. The memory stores successful actions for each item, but did not record failures. This modification aligns the memory mechanism with SC and DECKARD baselines, enabling a more consistent comparison across

baselines in our experimental setup. Intrinsic goal selection is random, as we do not assume such an expert-defined exploration curriculum.

**RAND.** RAND is a simple baseline specifically designed for our experimental setup. It started with an empty initial inventory and an LLM-predicted requirement set for each item. RAND did not incorporate any action memory. Its intrinsic goal selection involved randomly selecting from unexperienced items.

## J.2 MineRL environment

### J.2.1 Basic rules

Minecraft has been adopted as a suitable testbed for validating performance of AI agents on long-horizon tasks (Mao et al., 2022; Lin et al., 2021; Baker et al., 2022; Li et al., 2025a), largely because of the inherent dependency in item acquisition where agents must obtain prerequisite items before more advanced ones. Specifically, Minecraft features multiple technology levels—including wood, stone, iron, gold, diamond, etc. —which dictate item and tool dependencies. For instance, an agent must first craft a lower-level tool like a wooden pickaxe to mine materials such as stone. Subsequently, a stone pickaxe is required to mine even higher-level materials like iron. An iron pickaxe is required to mine materials like gold and diamond. Respecting the dependency is crucial for achieving complex goals, such as crafting an iron sword or mining a diamond.

### J.2.2 Observation and action space

First, we employ MineRL (Guss et al., 2019) with Minecraft version 1.16.5.

**Observation.** When making a plan, our agent receives inventory information (i.e., item with their quantities) as text. When executing the plan, our agent receives an RGB image with dimensions of $640 \times 360$, including the hotbar, health indicators, food saturation, and animations of the player's hands.

**Action space.** Following Optimus-1 (Li et al., 2024b), our low-level action space primarily consists of keyboard and mouse controls, except for craft and smelt high-level actions. Crucially, craft and smelt actions are included into our action space, following (Li et al., 2024b). This means these high-level actions automatically succeed in producing an item if the agent possesses all the required items and a valid actions for that item is chosen; otherwise, they fail. This abstraction removes the need for complex, precise low-level mouse control for these specific actions. For low-level controls, keyboard presses control agent movement (e.g., jumping, moving forward, backward) and mouse movements control the agent's perspective. The mouse's left and right buttons are used for attacking, using, or placing items. The detailed action space is described in Table 9.

Table 9: Action space in MineRL environment

| Index | Action | Human Action | Description |
|-------|--------|--------------|-------------|
| 1 | Forward | key W | Move forward. |
| 2 | Back | key S | Move back. |
| 3 | Left | key A | Move left. |
| 4 | Right | key D | Move right. |
| 5 | Jump | key Space | Jump. When swimming, keeps the player afloat. |
| 6 | Sneak | key left Shift | Slowly move in the current direction of movement. |
| 7 | Sprint | key left Ctrl | Move quickly in the direction of current movement. |
| 8 | Attack | left Button | Destroy blocks (hold down); Attack entity (click once). |
| 9 | Use | right Button | Place blocks, entity, open items or other interact actions defined by game. |
| 10 | hotbar [1-9] | keys 1-9 | Selects the appropriate hotbar item. |
| 11 | Open/Close Inventory | key E | Opens the Inventory. Close any open GUI. |
| 12 | Yaw | move Mouse X | Turning; aiming; camera movement.Ranging from -180 to +180. |
| 13 | Pitch | move Mouse Y | Turning; aiming; camera movement.Ranging from -180 to +180. |
| 14 | Craft | - | Execute crafting to obtain new item |
| 15 | Smelt | - | Execute smelting to obtain new item. |

### J.2.3 GOALS

We consider 67 goals from the long-horizon tasks benchmark suggested in (Li et al., 2024b). These goals are categorized into 7 groups based on Minecraft's item categories: Wood 🪵, Stone 🪨, Iron 🪙, Gold 🟨, Diamond 💎, Redstone 🔴, and Armor 🗡. All goal items within each group are listed in Table 10.

Table 10: Setting of 7 groups encompassing 67 Minecraft long-horizon goals.

| Group | Goal Num. | All goal items |
|---|---|---|
| 🪵 Wood | 10 | bowl, crafting_table, chest, ladder, stick, wooden_axe, wooden_hoe, wooden_pickaxe, wooden_shovel, wooden_sword |
| 🪨 Stone | 9 | charcoal, furnace, smoker, stone_axe, stone_hoe, stone_pickaxe, stone_shovel, stone_sword, torch |
| 🪙 Iron | 16 | blast_furnace, bucket, chain, hopper, iron_axe, iron_bars, iron_hoe, iron_nugget, iron_pickaxe, iron_shovel, iron_sword, rail, shears, smithing_table, stonecutter, tripwire_hook |
| 🟨 Gold | 6 | gold_ingot, golden_axe, golden_hoe, golden_pickaxe, golden_shovel, golden_sword |
| 🔴 Redstone | 6 | activator_rail, compass, dropper, note_block, piston, redstone_torch |
| 💎 Diamond | 7 | diamond, diamond_axe, diamond_hoe, diamond_pickaxe, diamond_shovel, diamond_sword, jukebox |
| 🗡 Armor | 13 | diamond_boots, diamond_chestplate, diamond_helmet, diamond_leggings, golden_boots, golden_chestplate, golden_helmet, golden_leggings, iron_boots, iron_chestplate, iron_helmet, iron_leggings, shield |

**Additional goals for scalability experiments..** To evaluate the scalability of XENON with respect to the number of goals Appendix K.9, we extend the above 67-goal set (Table 10) by adding additional goal items to construct two larger settings with 100 and 120 goals; the added goals are listed in Table 11.

Specifically, in the setting with 100 goals, we add 33 goals in total by introducing new "leather", "paper", and "flint" groups and by adding more items to the existing "wood" and "stone" groups. In the setting with 120 goals, we further add 20 goals in the "iron", "gold", "redstone", and "diamond" groups.

### J.2.4 EPISODE HORIZON

The episode horizon varies depending on the experiment phase: dependency learning or long-horizon goal planning. During the dependency learning phase, each episode has a fixed horizon of 36,000 steps. In this phase, if the agent successfully achieves an intrinsic goal within an episode, it is allowed to select another intrinsic goal and continue exploration without the episode ending. After dependency learning, when measuring the success rate of goals from the long-horizon task benchmark, the episode horizon differs based on the goal's category group. And in this phase, the episode immediately terminates upon success of a goal. The specific episode horizons for each group are as follows: Wood: 3,600 steps; Stone: 7,200 steps; Iron: 12,000 steps; and Gold, Diamond, Redstone, and Armor: 36,000 steps each.

### J.2.5 ITEM SPAWN PROBABILITY DETAILS

Following Optimus-1's public implementation, we have modified environment configuration different from original MineRL environment (Guss et al., 2019). In Minecraft, obtaining essential resources

Table 11: Additional goals used for the scalability experiments. The setting with 100 goals extends the 67-goal set in Table 10 by adding all items in the top block; the setting with 120 goals further includes both the top and bottom blocks.

| Group | Goal Num. | Added goal items |
|---|---|---|
| *Additional items in the setting with 100 goals (33 items)* | | |
| leather | 7 | leather, leather_boots, leather_chestplate, leather_helmet, leather_leggings, leather_horse_armor, item_frame |
| paper | 5 | map, book, cartography_table, bookshelf, lectern |
| flint | 4 | flint, flint_and_steel, fletching_table, arrow |
| wood | 8 | bow, boat, wooden_slab, wooden_stairs, wooden_door, wooden_sign, wooden_fence, woodenfence_gate |
| stone | 9 | cobblestone_slab, cobblestone_stairs, cobblestone_wall, lever, stone_slab, stone_button, stone_pressure_plate, stone_bricks, grindstone |
| *Additional items only in the setting with 120 goals (20 more items)* | | |
| iron | 7 | iron_trapdoor, heavy_weighted_pressure_plate, iron_door, crossbow, minecart, cauldron, lantern |
| gold | 4 | gold_nugget, light_weighted_pressure_plate, golden_apple, golden_carrot |
| redstone | 7 | redstone, powered_rail, target, dispenser, clock, repeater, detector_rail |
| diamond | 2 | obsidian, enchanting_table |

such as iron, gold, and diamond requires mining their respective ores. However, these ores are naturally rare, making them challenging to obtain. This inherent difficulty can significantly hinder an agent's goal completion, even with an accurate plan. This challenge in resource gathering due to an imperfect controller is a common bottleneck, leading many prior works to employ environmental modifications to focus on planning. For example, DEPS (Wang et al., 2023b) restricts the controller's actions based on the goal items [4]. Optimus-1 (Li et al., 2024b) also made resource items easier to obtain by increasing item ore spawn probabilities. To focus on our primary goal of robust planning and isolate this challenge, we follow Optimus-1 and adopt its item ore spawn procedure directly from the publicly released Optimus-1 repository, without any modifications to its source code [5].

The ore spawn procedure probabilistically spawns ore blocks in the vicinity of the agent's current coordinates $(x, y, z)$. Specifically, at each timestep, the procedure has a 10% chance of activating. When activated, it spawns a specific type of ore block based on the agent's y-coordinate. Furthermore, for any given episode, the procedure is not activate more than once at the same y-coordinate. The types of ore blocks spawned at different y-levels are as follows:

- *Coal Ore*: between y=45 and y=50.
- *Iron Ore*: between y=26 and y=43.
- *Gold Ore*: between y=15 and y=26
- *Redstone Ore*: between y=15 and y=26
- *Diamond Ore*: below y=14

---

[4]https://github.com/CraftJarvis/MC-Planner/blob/main/controller.py
[5]https://github.com/JiuTian-VL/Optimus-1/blob/main/src/optimus1/env/wrapper.py

### J.3 MINEFLAYER ENVIRONMENT

We use the Mineflayer (PrismarineJS, 2023) environment with Minecraft version 1.19. In Mineflayer, resource item spawn probabilities do not need to be adjusted, unlike in MineRL Appendix J.2.5. This is because the controller, JavaScript APIs provided by Mineflayer, is competent to gather many resource items.

#### J.3.1 OBSERVATION AND ACTION SPACE

The agent's observation space is multimodal. For planning, the agent receives its current inventory (i.e., item names and their quantities) as text. For plan execution, it receives a first-person RGB image that includes the hotbar, health and food indicators, and player hand animations. For action space, following ADAM (Yu & Lu, 2024), we use the JavaScript APIs provided by Mineflayer for low-level control. Specifically, our high-level actions, such as "craft", "smelt", and "mine", are mapped to corresponding Mineflayer APIs like `craftItem`, `smeltItem`, and `mineBlock`.

#### J.3.2 EPISODE HORIZON

For dependency learning, each episode has a fixed horizon of 30 minutes, which is equivalent to 36,000 steps in the MineRL environment. If the agent successfully achieves a goal within this horizon, it selects another exploratory goal and continues within the same episode.

### J.4 MC-TEXTWORLD

MC-Textworld is a text-based environment based on Minecraft game rules (Zheng et al., 2025). We employ Minecraft version 1.16.5. In this environment, basic rules and goals are the same as those in the MineRL environment Appendix J.2. Furthermore, resource item spawn probabilities do not need to be adjusted, unlike in MineRL Appendix J.2.5. This is because an agent succeeds in mining an item immediately without spatial exploration, if it has a required tool and "mine" is a valid action for that item.

In the following subsections, we detail the remaining aspects of experiment setups in this environment: the observation and action space, and the episode horizon.

#### J.4.1 OBSERVATION AND ACTION SPACE

The agent receives a text-based observation consisting of inventory information (i.e., currently possessed items and their quantities). Actions are also text-based, where each action is represented as an high-level action followed by an item name (e.g., "mine diamond"). Thus, to execute a subgoal specified as $(a, q, v)$ (high-level action $a$, quantity $q$, item $v$), the agent repeatedly performs the action $(a, v)$ until $q$ units of $v$ are obtained.

#### J.4.2 EPISODE HORIZON

In this environment, we conduct experiments for dependency learning only. Each episode has a fixed horizon of 3,000 steps. If the agent successfully achieves an intrinsic goal within an episode, it is then allowed to select another intrinsic goal and continue exploration, without termination of the episode.

#### J.4.3 PERTURBATION ON GROUND TRUTH RULES

To evaluate each agent's robustness to conflicts with its prior knowledge, we perturb the ground-truth rules (required items and actions) for a subset of goal items, as shown in Figure 21. The perturbation is applied at different intensity levels (from 1 to 3), where higher levels affect a greater number of items. These levels are cumulative, meaning a Level 2 perturbation includes all perturbations from Level 1 plus additional ones.

- **Vanilla Setting**: In the setting with no perturbation (Figure 21, a), the ground-truth rules are unmodified. In the figure, items in the black solid boxes are the goal items, and those with arrows pointing to them are their true required items. Each goal item has "craft" as a valid action.

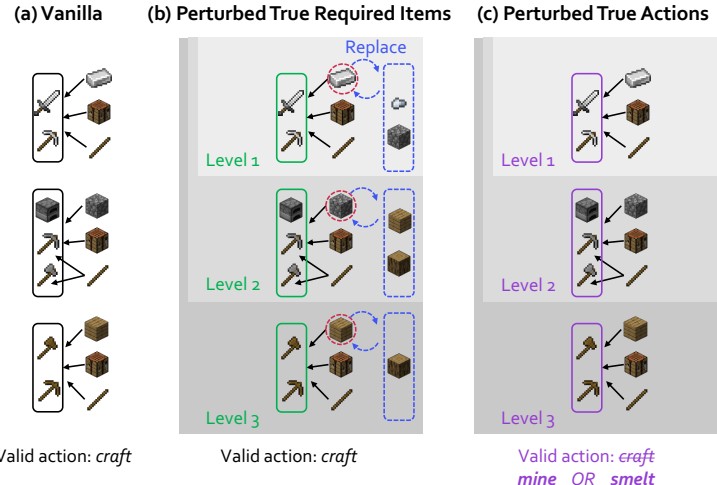

Figure 21: Illustration of the ground-truth rule perturbation settings. (a) in the vanilla setting, goal items (black boxes) have standard required items (incoming edges) and "craft" is the valid action; (b) in the Perturbed Requirements setting, one required item (red dashed circle) is replaced by a new one randomly from a candidate pool (blue dashed box); (c) in the Perturbed Actions setting, the valid action is changed to either "mine" or "smelt".

- **Perturbed True Required Items**: In this setting (Figure 21, b), one of the true required items (indicated by a red dashed circle) for a goal is replaced. The new required item is chosen uniformly at random from a candidate pool (blue dashed box). The valid action remains craft.
- **Perturbed True Actions**: In this setting (Figure 21, c), the valid action for a goal is randomly changed from "craft" to either "mine" or "smelt". The required items are not modified.
- **Perturbed Both Rules**: In this setting, both the required items and the valid actions are modified according to the rules described above.

# K  ADDITIONAL EXPERIMENTAL RESULTS

## K.1  LLM-PREDICTED INITIAL DEPENDENCY GRAPH ANALYSIS

Table 12: Performance analysis for the initial LLM-predicted requirement sets over 75 Minecraft items, used to build the initial dependency graph. Note that while we began the prediction process with 67 goal items, the total number of predicted items expanded to 75. This expansion occurred because, as the LLM predicted requirement sets for items in the dependency graph (initially for those goal items), any newly mentioned items that were not yet part of the graph are also included. This iterative process is detailed in Section 4.1 (Dependency graph initialization) of our method.

| Metric | Value |
|---|---|
| *Requirement Set Prediction Accuracy* | |
| Correct items (ignoring quantities) | 23% |
| Exact items & quantities | 8% |
| *Non-existent Item Rates* | |
| Non-existent items | 8% |
| Descendants of non-existent items | 23% |
| *Required Items Errors* | |
| Unnecessary items included | 57% |
| Required items omitted | 57% |
| *Required Item Quantity Prediction Errors* | |
| Standard deviation of quantity error | 2.74 |
| Mean absolute quantity error | 2.05 |
| Mean signed quantity error | -0.55 |

The initial dependency graph, constructed from predictions by Qwen2.5-VL-7B (Bai et al., 2025), forms the initial planning knowledge for XENON (Section 4.1). This section analyzes its quality, highlighting limitations that necessitate our adaptive dependency learning.

As shown in Table 12, the 7B LLM's initial requirement sets exhibit significant inaccuracies. Accuracy for correct item types was 23%, dropping to 8% for exact items and quantities. Errors in dependency among items are also prevalent: 57% of items included unnecessary items, and 57% omitted required items. Furthermore, 8% of predicted items were non-existent (hallucinated), making 23% of descendant items unattainable. Quantity predictions also showed substantial errors, with a mean absolute error of 2.05.

These results clearly demonstrate that the LLM-generated initial dependency graph is imperfect. Its low accuracy and high error rates underscore the unreliability of raw LLM knowledge for precise planning, particularly for smaller models like the 7B LLM which are known to have limited prior knowledge on Minecraft, as noted in previous work (ADAM, Yu & Lu (2024), Appendix A. LLMs' Prior Knowledge on Minecraft). This analysis therefore highlights the importance of the adaptive dependency learning within XENON, which is designed to refine this initial, imperfect knowledge for robust planning.

Table 13: Ratio of dependencies learned for items which are unobtainable by the flawed initial dependency graph (out of 51). Analysis is based on the final learned graphs from the MineRL experiments.

| Agent | Learned ratio (initially unobtainable items) |
|---|---|
| XENON | 0.51 |
| SC | 0.25 |
| DECKARD | 0.25 |
| ADAM | 0.00 |
| RAND | 0.02 |

### K.2 ADDITIONAL ANALYSIS OF LEARNED DEPENDENCY GRAPH

As shown in Table 13, XENON demonstrates significantly greater robustness to the LLM's flawed prior knowledge compared to all baselines. It successfully learned the correct dependencies for over half (0.51) of the 51 items that were initially unobtainable by the flawed graph. In contrast, both DECKARD (with no correction) and the SC baseline (with LLM self-correction) learned only a quarter of these items (0.25). This result strongly indicates that relying on the LLM to correct its own errors is as ineffective as having no correction mechanism at all in this setting. The other baselines, ADAM and RAND, failed almost completely, highlighting the difficulty of this challenge.

### K.3 IMPACT OF CONTROLLER CAPACITY ON DEPENDENCY LEARNING

We observe that controller capacity significantly impacts an agent's ability to learn dependencies from interaction. Specifically, in our MineRL experiments, we find that ADAM fails to learn any new dependencies due to the inherent incompatibility between its strategy and the controller's limitations. In our realistic setting with empty initial inventories, ADAM's strategy requires gathering a sufficient quantity (fixed at 8, same with our hyperparameter $\alpha_i$[6]) of all previously used resources before attempting a new item. This list of required resource items includes gold ingot 🪙, because of an initially provided human-written plan for golden sword; however, the controller STEVE-1 never managed to collect more than seven units of gold in a single episode across all our experiments. Consequently, this controller bottleneck prevents ADAM from ever attempting to learn new items, causing its dependency learning to stall completely.

Although XENON fails to learn dependencies for the Redstone group items in MineRL, our analysis shows this stems from controller limitations rather than algorithmic ones. Specifically, in MineRL, STEVE-1 cannot execute XENON's exploration strategy for *inadmissible items*, which involves gathering a sufficient quantity of all previously used resources before a retry (Section 4.1). The Redstone group items become inadmissible because the LLM's initial predictions for them are entirely incorrect. This lack of a valid starting point prevents XENON from ever experiencing the core item, redstone, being used as a requirement for any other item. Consequently, our `RevisionByAnalogy` mechanism has no analogous experience to propose redstone as a potential required item for other items during its revision process.

In contrast, with more competent controllers, XENON successfully overcomes even such severely flawed prior knowledge to learn the challenging Redstone group dependencies, as demonstrated in Mineflayer and MC-TextWorld. First, in Mineflayer, XENON learns the correct dependencies for 5 out of 6 Redstone items. This success is possible because its more competent controller can execute the exploration strategy for *inadmissible items*, which increases the chance of possessing the core required item (redstone) during resource gathering. Second, with a perfect controller in MC-TextWorld, XENON successfully learns the dependencies for all 6 Redstone group items in every single episode.

### K.4 IMPACT OF CONTROLLER CAPACITY IN LONG-HORIZON GOAL PLANNING

Because our work focuses on building a robust planner, to isolate the planning from the significant difficulty of item gathering—a task assigned to the controller—our main experiments for long-horizon tasks (Section 5.3) uses a modified MineRL environment following the official implementation of Optimus-1. This modification makes essential resource items like iron, gold, and diamond easier for the controller to find, allowing for a clearer evaluation of planning algorithms (modifications are detailed in Appendix J.2.5). However, to provide a more comprehensive analysis, we also evaluated our agent and baselines in the unmodified, standard MineRL environment. In this setting, items like iron, gold, and diamond are naturally rare, making item gathering a major bottleneck.

The results are shown in Table 14. Most importantly, XENON* consistently outperforms the baselines in both the modified and standard MineRL. Notably, in the standard environment, XENON*'s performance on the Iron group (0.24 SR) is comparable to that of the *OracleActionPlanner* (0.27 SR), which always generates correct plans for all goals. This comparison highlights the severity of the controller bottleneck: even the *OracleActionPlanner* achieves a 0.00 success rate for the Diamond

---

[6]The scaling factor for required item quantities for inadmissible items.

Table 14: Long-horizon task success rate (SR) comparison between the **Modified MineRL** (a setting where resource items are easier to obtain) and **Standard MineRL** environments. All methods are provided with the correct dependency graph. DEPS† and Optimus-1† are our reproductions of the respective methods using Qwen2.5-VL-7B as a planner. *OracleActionPlanner*, which generates the correct plan for all goals, represents the performance upper bound. SR for Optimus-1† and XENON* in the **Modified MineRL** column are taken from Table 3 in Section 5.3.

| Method | Dependency | Modified MineRL | | | Standard MineRL | | |
|---|---|---|---|---|---|---|---|
| | | Iron | Diamond | Gold | Iron | Diamond | Gold |
| DEPS† | - | 0.02 | 0.00 | 0.01 | 0.01 | 0.00 | 0.00 |
| Optimus-1† | Oracle | 0.23 | 0.10 | 0.11 | 0.13 | 0.00 | 0.00 |
| XENON* | Oracle | **0.83** | **0.75** | **0.73** | **0.24** | **0.00** | **0.00** |
| *OracleActionPlanner* | Oracle | - | - | - | *0.27* | *0.00* | *0.00* |

and Gold groups in the standard MineRL. This shows that the failures are due to the controller's inability to gather rare resources in the standard environment.

### K.5 LONG-HORIZON TASK BENCHMARK EXPERIMENTS ANALYSIS

This section provides a detailed analysis of the performance differences observed in Table 3 between Optimus-1† and XENON* on long-horizon tasks, even when both access to a true dependency graph and increased item spawn probabilities (Appendix J.2.5). We specifically examine various plan errors encountered when reproducing Optimus-1† using Qwen2.5-VL-7B as the planner, and explain how XENON* robustly constructs plans through step-by-step planning with FAM.

Table 15: Analysis of primary plan errors observed in Optimus-1† and XENON* during long-horizon tasks benchmark experiments. This table presents the ratio of specified plan error among the failed episodes for Optimus-1† and XENON* respectively. *Invalid Action* indicates errors where an invalid action is used for an item in a subgoal. *Subgoal Omission* refers to errors where a necessary subgoal for a required item is omitted from the plan. Note that these plan error values are not exclusive; one episode can exhibit multiple types of plan errors.

| Plan Error Type | Optimus-1† Error Rate (%) | XENON* Error Rate (%) |
|---|---|---|
| Invalid Action | 37 | 2 |
| Subgoal Omission | 44 | 0 |

Optimus-1† has no fine-grained action knowledge correction mechanism. Furthermore, Optimus-1†'s LLM planner generates a long plan at once with a long input prompt including a sequence of aggregated requirements $((q_1, u_1), \ldots, (q_{L_v}, u_{L_v}) = (1, v))$ for the goal item $v$. Consequently, as shown in Table 15, Optimus-1 generates plans with invalid actions for required items, denoted as Invalid Action. Furthermore, Optimus-1 omits necessary subgoals for required items, even they are in the input prompts, denoted as Subgoal Omission.

In contrast, XENON discovers valid actions by leveraging FAM, which records the outcomes of each action for every item, thereby enabling it to avoid empirically failed ones and and reuse successful ones. Furthermore, XENON mitigates the problem of subgoal omission through constructing a plan by making a subgoal for each required item one-by-one.

## K.6 ROBUST DEPENDENCY LEARNING UNDER DYNAMIC TRUE KNOWLEDGE

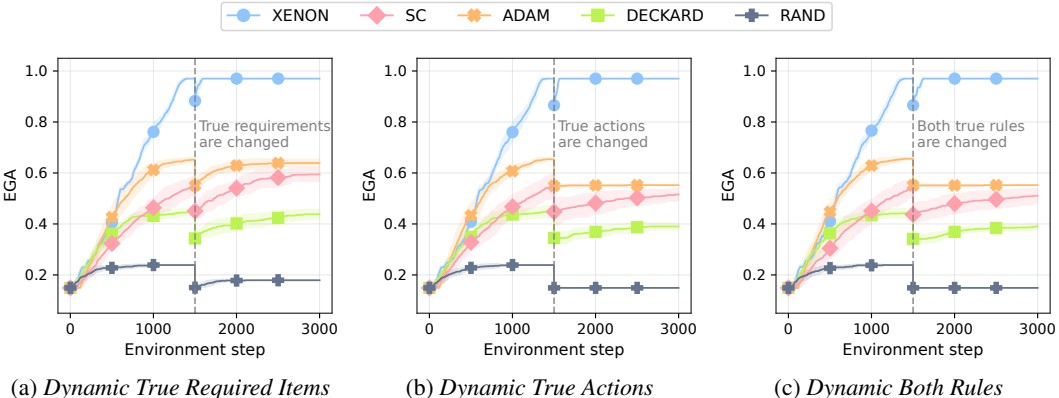

Figure 22: **Robustness against dynamic true knowledge.** EGA over 3,000 environment steps in the where the true item acquisition rules are changed during the learning process.

Additionally, We show XENON is also applicable to scenarios where the latent true knowledge changes dynamically. We design three dynamic scenarios where the environment begins with the vanilla setting, `(0,0)`, for the first 1,500 steps, then transitions to a level-3 perturbation setting for the subsequent 1,500 steps: either required items-only `(3,0)`, action-only `(0,3)`, or both `(3,3)`. Upon this change, the agent is informed of which items' rules are modified but not what the new rules are, forcing it to relearn from experience. As shown in Figure 22, XENON rapidly adapts by re-learning the new dependencies and recovering its near-perfect EGA in all three scenarios. In contrast, all baselines fail to adapt effectively, with their performance remaining significantly degraded after the change. Specifically, for the 7 items whose rules

Table 16: The ratio of correctly learned dependencies among whose rules are dynamically changed (out of 7 total) by each agent. Columns correspond to type of ground-truth rules changed during learning: requirements only, actions only, or both.

| Agent | (3,0) | (0,3) | (3,3) |
|---|---|---|---|
| XENON | **1.0** | **1.0** | **1.0** |
| SC | 0.80 | 0.0 | 0.0 |
| ADAM | 0.83 | 0.0 | 0.0 |
| DECKARD | 0.49 | 0.0 | 0.0 |
| RAND | 0.29 | 0.0 | 0.0 |

are altered, Table 16 shows that XENON achieves a perfect re-learning ratio of 1.0 in all scenarios, while all baselines score 0.0 whenever actions are modified.

## K.7 ABLATION STUDIES FOR LONG-HORZION GOAL PLANNING

Table 17: Ablation experiment results for long-horizon goal planning in MineRL. Without Learned Dependency, XENON employs a dependency graph initialized with LLM predictions and human-written examples. Without Action Correction, XENON saves and reuses successful actions in FAM, but it does not utilize the information of failed actions.

| Learned Dependency | Action Correction | CRe | Wood | Stone | Iron | Diamond | Gold | Armor | Redstone |
|---|---|---|---|---|---|---|---|---|---|
| | | | 0.54 | 0.39 | 0.10 | 0.26 | 0.45 | 0.0 | 0.0 |
| | ✓ | | 0.54 | 0.38 | 0.09 | 0.29 | 0.45 | 0.0 | 0.0 |
| ✓ | | | 0.82 | 0.69 | 0.36 | 0.59 | 0.69 | 0.22 | 0.0 |
| ✓ | ✓ | | 0.82 | 0.79 | 0.45 | 0.59 | 0.68 | 0.21 | 0.0 |
| ✓ | ✓ | ✓ | **0.85** | **0.81** | **0.46** | **0.64** | **0.74** | **0.28** | 0.0 |

To analyze how each of XENON's components contributes to its long-horizon planning, we conducted an ablation study in MineRL, with results shown in Table 17. The findings first indicate that without accurate dependency knowledge, our action correction using FAM provides no significant benefit on its own (row 1 vs. row 2). The most critical component is the learned dependency graph, which dramatically improves success rates across all item groups (row 3). Building on this, adding FAM's

action correction further boosts performance, particularly for the Stone and Iron groups where it helps overcome the LLM's flawed action priors (row 4). Finally, Context-aware Reprompting (CRe, Section 4.3) provides an additional performance gain on more challenging late-game items, such as Iron, Gold, and Armor. This is likely because their longer episode horizons offer more opportunities for CRe to rescue a stalled controller.

## K.8  The Necessity of Knowledge Correction even with External Sources

Even when an external source is available to initialize an agent's knowledge, correcting that knowledge from interaction remains essential for dependency and action learning, because such sources can be flawed or outdated. To support this, we evaluate XENON and the baselines in the MC-TextWorld environment where each agent's dependency graph is initialized from an oracle graph, while the environment's ground-truth dependency graph is perturbed (perturbation level 3 in Table 4). We measure performance as the ratio of the 67 goal items obtained within a single episode. We use an intrinsic exploratory item selection method for all agents (i.e., which item each agent chooses on its own to try to obtain next): they choose, among items not yet obtained in the current episode, the one with the fewest attempts so far.

As shown in Figure 23, this experiment demonstrates that, even when an external source is available, (1) interaction experience-based knowledge correction remains crucial when the external source is mismatched with the environment, and (2) XENON is also applicable and robust in this

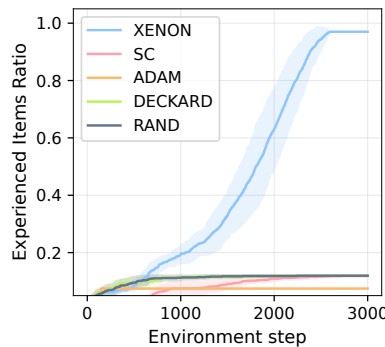

Figure 23: Ratio of goal items obtained in one MC-TextWorld episode when each agent's dependency graph is initialized from an oracle graph while the environment's ground-truth dependency graph is perturbed. Solid lines denote the mean over 15 runs; shaded areas denote the standard deviation.

scenario. By continually revising its dependency knowledge, XENON achieves a much higher ratio of goal items obtained in an episode than all baselines. In contrast, the baselines either rely on unreliable LLM self-correction (e.g., SC) or do not correct flawed knowledge at all (e.g., DECKARD, ADAM, RAND), and therefore fail to obtain many goal items. Their performance is especially poor because there are dependencies between goals: for example, when the true required items for stone pickaxe and iron pickaxe are perturbed, the baselines cannot obtain these items and thus cannot obtain other goal items that depend on them.

## K.9  Scalability of Dependency and Action Learning with More Goals and Actions

To evaluate the scalability of XENON's dependency and action learning, we vary the number of goal items and available actions in the MC-TextWorld environment. For the goal-scaling experiment, we increase the number of goals from 67 to 100 and 120 by adding new goal items (see Table 11 for the added goals), while keeping the original three actions "mine", "craft", and "smelt" fixed. For the action-scaling experiment, we increase the available actions from 3 to 15, 30, and 45 (e.g., "harvest", "hunt", "place"), while keeping the original 67 goals fixed.

The results in Figure 24 show that XENON maintains high EGA as both the number of goals and the number of actions grow, although the number of environment steps required for convergence naturally increases. As seen in Figure 24a, increasing the number of goals from 67 to 100 and 120 only moderately delays convergence (from around 1,400 to about 2,100 and 2,600 steps). In contrast, Figure 24b shows a larger slowdown when increasing the number of actions (from about 1,400 steps with 3 actions to roughly 4,000, 7,000, and 10,000 steps with 15, 30, and 45 actions), which is expected because XENON only revises an item's dependency after all available actions for that item have been classified as empirically invalid by FAM. We believe this convergence speed could be improved with minimal changes, such as by lowering $x_0$, the failure count threshold for classifying an action as invalid, or by triggering dependency revision once the agent has failed to obtain an item a fixed number of times, regardless of which actions were tried in subgoals.

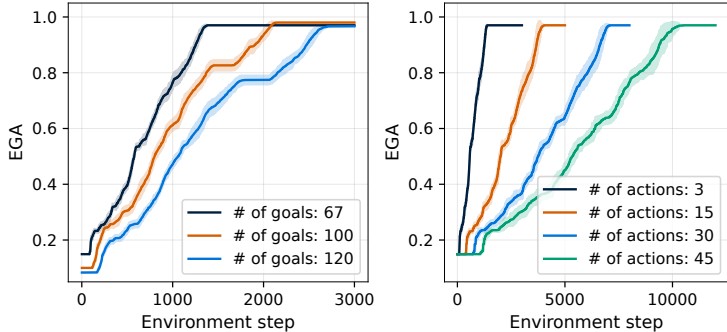

(a) Effect of increasing the number of goals

(b) Effect of increasing the number of actions

Figure 24: **Scalability of XENON with more goals and actions.** EGA over environment steps in MC-TextWorld when (a) increasing the number of goal items and (b) increasing the number of available actions. In (a), we keep the three actions ("mine", "craft", "smelt") fixed, while in (b) we keep the 67 goal items fixed. Solid lines denote the mean over 15 runs; shaded areas denote the standard deviation.

### K.10 ABLATION ON ACTION SELECTION METHODS FOR MAKING SUBGOALS

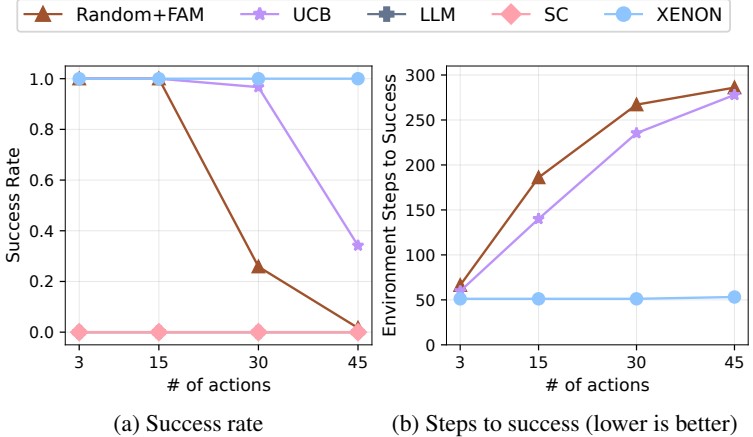

(a) Success rate

(b) Steps to success (lower is better)

Figure 25: **Ablation on action selection methods for subgoal construction.** We evaluate different action selection methods for solving long-horizon goals given an oracle dependency graph, as the size of the available action set increases. (a) Success rate and (b) number of environment steps per successful episode. Note that in (a), the curves for LLM and SC overlap at 0.0 because they fail on all episodes, and in (b), they are omitted since they never succeed.

We find that, while LLMs can in principle accelerate the search for valid actions, they do so effectively *only when their flawed knowledge is corrected algorithmically*. To support this, we study how different action selection methods for subgoal construction affect performance on long-horizon goals. In this ablation, the agent is given an oracle dependency graph and a long-horizon goal, and only needs to output one valid action from the available actions for each subgoal item to achieve that goal. Each episode specifies a single goal item, and it is counted as successful if the agent obtains this item within 300 environment steps in MC-TextWorld. To study scalability with respect to the size of the available action set, we vary the number of actions as 3, 15, 30, and 45 by gradually adding actions such as "harvest" and "hunt" to the original three actions ("mine", "craft", "smelt").

**Methods and metrics.** We compare five action selection methods: **Random+FAM** (which randomly samples from available actions that have not yet repeatedly failed and reuses past successful actions),

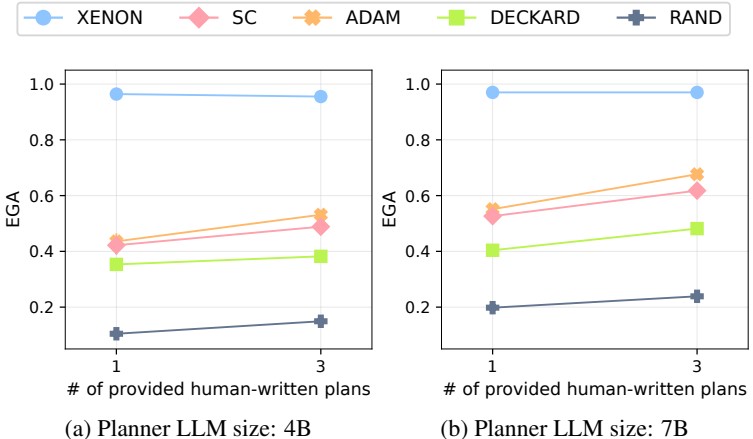

(a) Planner LLM size: 4B      (b) Planner LLM size: 7B

Figure 26: **Effect of planner LLM size and initial dependency graph quality in dependency and action learning.** The plots show EGA after 3,000 environment steps of dependency and action learning in MC-TextWorld, obtained by varying the planner LLM size and the amount of correct knowledge in the initial dependency graph (controlled by the number of provided human-written plans). In (a), the planner is Phi-4-mini (4B) (Microsoft et al., 2025); in (b), the planner is Qwen2.5-VL-7B (7B) (Bai et al., 2025).

**UCB**, **LLM** without memory, LLM self-correction (**SC**), and **XENON**, which combines an LLM with FAM. We report the average success rate and the average number of environment steps to success over 20 runs per goal item, where goal items are drawn from the Redstone group.

As shown in Figure 25, among the three LLM-based methods (LLM, SC, XENON), only XENON—which corrects the LLM's action knowledge by removing repeatedly failed actions from the set of candidate actions the LLM is allowed to select—solves long-horizon goals reliably, maintaining a success rate of 1.0 and requiring roughly 50 environment steps across all sizes of the available action set. In contrast, LLM and SC never succeed in any episode, because they keep selecting incorrect actions for subgoal items (e.g., redstone), and therefore perform worse than the non-LLM baselines, Random+FAM and UCB. Random+FAM and UCB perform well when the number of available actions is small, but become increasingly slow and unreliable as the number of actions grows, often failing to reach the goal within the episode horizon.

## K.11 ROBUSTNESS TO SMALLER PLANNER LLMS AND LIMITED INITIAL KNOWLEDGE

We further evaluate robustness of XENON and the baselines to limited prior knowledge by measuring dependency and action learning in MC-TextWorld while (i) varying the planner LLM size and (ii) degrading the quality of the initial dependency graph. For the planner LLM, we compare a 7B model (Qwen2.5-VL-7B (Bai et al., 2025)) against a 4B model (Phi-4-mini (Microsoft et al., 2025)); for the initial graph quality, we vary the number of provided human-written plans used to initialize the graph from three ("craft iron_sword", "mine diamond", "craft golden_sword") to one ("craft iron_sword").

As shown in Figure 26, XENON remains robust across all these settings: its EGA stays near-perfect even with the smaller 4B planner and the weakest initial graph, indicating that leveraging experiences can quickly compensate for weak priors. In contrast, baselines that rely on LLM self-correction (SC) or that strongly depend on the LLM or initial graph (ADAM, DECKARD) suffer substantial drops in EGA as the planner LLM becomes smaller and the initial graph contains less correct prior knowledge. This suggests that, in our setting, algorithmic knowledge correction is more critical than scaling up the planner LLM or richer initial human-provided knowledge.

### K.12 FULL RESULTS ON THE LONG-HORIZON TASKS BENCHMARK

In this section, we report XENON's performance on each goal within the long-horizon tasks benchmark, detailing metrics such as the goal item, number of sub-goals, success rate (SR), and evaluation episodes.

Table 18 and 19 present XENON's results when utilizing the dependency graph learned through 400 episodes of exploration. Conversely, Table 20 and 21 display XENON*'s performance, which leverages an oracle dependency graph.

Table 18: The results of XENON (with dependency graph learned via exploration across 400 episodes) on the Wood group, Stone group, and Iron group. SR denotes success rate.

| Group | Goal | Sub-Goal Num. | SR | Eval Episodes |
|---|---|---|---|---|
| Wood | bowl | 4 | 92.68 | 41 |
| | chest | 4 | 95.24 | 42 |
| | crafting_table | 3 | 95.83 | 48 |
| | ladder | 5 | 0.00 | 31 |
| | stick | 3 | 95.45 | 44 |
| | wooden_axe | 5 | 90.91 | 44 |
| | wooden_hoe | 5 | 95.35 | 43 |
| | wooden_pickaxe | 5 | 93.02 | 43 |
| | wooden_shovel | 5 | 93.75 | 48 |
| | wooden_sword | 5 | 95.35 | 43 |
| Stone | charcoal | 8 | 87.50 | 40 |
| | furnace | 7 | 88.10 | 42 |
| | smoker | 8 | 0.00 | 47 |
| | stone_axe | 7 | 97.78 | 45 |
| | stone_hoe | 7 | 90.70 | 43 |
| | stone_pickaxe | 7 | 95.45 | 44 |
| | stone_shovel | 7 | 89.58 | 48 |
| | stone_sword | 7 | 89.80 | 49 |
| | torch | 7 | 93.02 | 43 |
| Iron | blast_furnace | 13 | 0.00 | 42 |
| | bucket | 11 | 0.00 | 47 |
| | chain | 12 | 0.00 | 42 |
| | hopper | 12 | 0.00 | 47 |
| | iron_axe | 11 | 75.56 | 45 |
| | iron_bars | 11 | 80.43 | 46 |
| | iron_hoe | 11 | 89.13 | 46 |
| | iron_nugget | 11 | 79.55 | 44 |
| | iron_pickaxe | 11 | 77.08 | 48 |
| | iron_shovel | 11 | 75.56 | 45 |
| | iron_sword | 11 | 84.78 | 46 |
| | rail | 11 | 0.00 | 44 |
| | shears | 11 | 0.00 | 43 |
| | smithing_table | 11 | 93.75 | 48 |
| | stonecutter | 12 | 0.00 | 43 |
| | tripwire_hook | 11 | 78.43 | 51 |

Table 19: The results of XENON (with dependency graph learned via exploration across 400 episodes) on the Gold group, Diamond group, Redstone group, and Armor group. SR denotes success rate.

| Group | Goal Item | Sub Goal Num. | SR | Eval Episodes |
|---|---|---|---|---|
| Gold | gold_ingot | 13 | 76.92 | 52 |
| | golden_axe | 14 | 72.00 | 50 |
| | golden_hoe | 14 | 66.67 | 48 |
| | golden_pickaxe | 14 | 76.00 | 50 |
| | golden_shovel | 14 | 71.74 | 46 |
| | golden_sword | 14 | 78.26 | 46 |
| Diamond | diamond | 12 | 87.76 | 49 |
| | diamond_axe | 13 | 72.55 | 51 |
| | diamond_hoe | 13 | 63.79 | 58 |
| | diamond_pickaxe | 13 | 60.71 | 56 |
| | diamond_shovel | 13 | 84.31 | 51 |
| | diamond_sword | 13 | 76.79 | 56 |
| | jukebox | 13 | 0.00 | 48 |
| Redstone | activator_rail | 14 | 0.00 | 3 |
| | compass | 13 | 0.00 | 3 |
| | dropper | 13 | 0.00 | 3 |
| | note_block | 13 | 0.00 | 4 |
| | piston | 13 | 0.00 | 12 |
| | redstone_torch | 13 | 0.00 | 19 |
| Armor | diamond_boots | 13 | 64.29 | 42 |
| | diamond_chestplate | 13 | 0.00 | 44 |
| | diamond_helmet | 13 | 67.50 | 40 |
| | diamond_leggings | 13 | 0.00 | 37 |
| | golden_boots | 14 | 69.23 | 39 |
| | golden_chestplate | 14 | 0.00 | 39 |
| | golden_helmet | 14 | 60.53 | 38 |
| | golden_leggings | 14 | 0.00 | 38 |
| | iron_boots | 11 | 94.44 | 54 |
| | iron_chestplate | 11 | 0.00 | 42 |
| | iron_helmet | 11 | 4.26 | 47 |
| | iron_leggings | 11 | 0.00 | 41 |
| | shield | 11 | 0.00 | 46 |

## K.13 EXPERIMENTS COMPUTE RESOURCES

All experiments were conducted on an internal computing cluster equipped with RTX3090, A5000, and A6000 GPUs. We report the total aggregated compute time from running multiple parallel experiments. For the dependency learning, exploration across 400 episodes in the MineRL environment, the total compute time was 24 days. The evaluation on the long-horizon tasks benchmark in the MineRL environment required a total of 34 days of compute. Experiments within the MC-TextWorld environment for dependency learning utilized a total of 3 days of compute. We note that these values represent aggregated compute time, and the actual wall-clock time for individual experiments was significantly shorter due to parallelization.

Table 20: The results of XENON* (with oracle dependency graph) on the Wood group, Stone group, and Iron group. SR denotes success rate.

| Group | Goal Item | Sub-Goal Num. | SR | Eval Episodes |
|---|---|---|---|---|
| Wood | bowl | 4 | 94.55 | 55 |
| | chest | 4 | 94.74 | 57 |
| | crafting_table | 3 | 94.83 | 58 |
| | ladder | 5 | 94.74 | 57 |
| | stick | 3 | 95.08 | 61 |
| | wooden_axe | 5 | 94.64 | 56 |
| | wooden_hoe | 5 | 94.83 | 58 |
| | wooden_pickaxe | 5 | 98.33 | 60 |
| | wooden_shovel | 5 | 96.49 | 57 |
| | wooden_sword | 5 | 94.83 | 58 |
| Stone | charcoal | 8 | 92.68 | 41 |
| | furnace | 7 | 90.00 | 40 |
| | smoker | 8 | 87.50 | 40 |
| | stone_axe | 7 | 95.12 | 41 |
| | stone_hoe | 7 | 94.87 | 39 |
| | stone_pickaxe | 7 | 94.87 | 39 |
| | stone_shovel | 7 | 94.87 | 39 |
| | stone_sword | 7 | 92.11 | 38 |
| | torch | 7 | 92.50 | 40 |
| Iron | blast_furnace | 13 | 82.22 | 45 |
| | bucket | 11 | 89.47 | 38 |
| | chain | 12 | 83.33 | 36 |
| | hopper | 12 | 77.78 | 36 |
| | iron_axe | 11 | 82.50 | 40 |
| | iron_bars | 11 | 85.29 | 34 |
| | iron_hoe | 11 | 75.68 | 37 |
| | iron_nugget | 11 | 84.78 | 46 |
| | iron_pickaxe | 11 | 83.33 | 42 |
| | iron_shovel | 11 | 78.38 | 37 |
| | iron_sword | 11 | 85.42 | 48 |
| | rail | 11 | 80.56 | 36 |
| | shears | 11 | 82.05 | 39 |
| | smithing_table | 11 | 83.78 | 37 |
| | stonecutter | 12 | 86.84 | 38 |
| | tripwire_hook | 11 | 91.18 | 34 |

## L  THE USE OF LARGE LANGUAGE MODELS (LLMS)

In preparing this manuscript, we used an LLM as a writing assistant to improve the text. Its role included refining grammar and phrasing, suggesting clearer sentence structures, and maintaining a consistent academic tone. All technical contributions, experimental designs, and final claims were developed by the human authors, who thoroughly reviewed and take full responsibility for the paper's content.

Table 21: The results of XENON* (with oracle dependency graph) on the Gold group, Diamond group, Redstone group, and Armor group. SR denotes success rate.

| Group | Goal Item | Sub Goal Num. | SR | Eval Episodes |
|---|---|---|---|---|
| Gold | gold_ingot | 13 | 78.38 | 37 |
| | golden_axe | 14 | 65.12 | 43 |
| | golden_hoe | 14 | 70.27 | 37 |
| | golden_pickaxe | 14 | 75.00 | 36 |
| | golden_shovel | 14 | 78.38 | 37 |
| Diamond | diamond | 12 | 71.79 | 39 |
| | diamond_axe | 13 | 70.00 | 40 |
| | diamond_hoe | 13 | 85.29 | 34 |
| | diamond_pickaxe | 13 | 72.09 | 43 |
| | diamond_shovel | 13 | 76.19 | 42 |
| | diamond_sword | 13 | 80.56 | 36 |
| | jukebox | 13 | 69.77 | 43 |
| Redstone | activator_rail | 14 | 67.39 | 46 |
| | compass | 13 | 70.00 | 40 |
| | dropper | 13 | 75.00 | 40 |
| | note_block | 13 | 89.19 | 37 |
| | piston | 13 | 65.79 | 38 |
| | redstone_torch | 13 | 84.85 | 33 |
| Armor | diamond_boots | 13 | 60.78 | 51 |
| | diamond_chestplate | 13 | 20.00 | 50 |
| | diamond_helmet | 13 | 71.79 | 39 |
| | diamond_leggings | 13 | 33.33 | 39 |
| | golden_boots | 14 | 75.00 | 40 |
| | golden_chestplate | 14 | 0.00 | 36 |
| | golden_helmet | 14 | 54.05 | 37 |
| | golden_leggings | 14 | 0.00 | 38 |
| | iron_boots | 11 | 93.62 | 47 |
| | iron_chestplate | 11 | 97.50 | 40 |
| | iron_helmet | 11 | 86.36 | 44 |
| | iron_leggings | 11 | 97.50 | 40 |
| | shield | 11 | 97.62 | 42 |

