# OpenReview forum: "Experience-based Knowledge Correction for Robust Planning in Minecraft"
_ICLR.cc/2026/Conference — ICLR 2026 Poster_

### Official Review · Reviewer_S6YL · 2025-10-23

**Soundness:** 2
**Presentation:** 3
**Contribution:** 2
**Rating:** 4
**Confidence:** 3

**Summary:**

This paper presents XENON (eXpErience-based kNOwledge correctioN), an LLM-based planning framework for embodied agents in long-horizon environments such as Minecraft. The paper argues that LLMs often carry flawed priors about item dependencies and actions, and fail to self-correct even with feedback. XENON introduces two algorithmic mechanisms—Adaptive Dependency Graph (ADG) for dependency correction and Failure-aware Action Memory (FAM) for action correction—allowing the agent to revise its external knowledge from binary success/failure feedback. Experiments in several Minecraft environments demonstrate that XENON achieves strong performance and robustness, outperforming prior agents including those relying on larger proprietary LLMs.

**Strengths:**

1. The paper is clearly written and easy to follow. The modular decomposition into ADG and FAM makes the system architecture understandable and well-motivated.

2. The implementation and experimental setup are extensive. The authors carefully evaluate across multiple environments (MineRL, Mineflayer, MC-TextWorld) and provide comprehensive ablations, demonstrating the effectiveness of each module.

3. The work highlights an important issue — how to make lightweight LLM-based agents robust to flawed priors and minimal feedback. This direction is relevant to building scalable, open-weight embodied agents.

**Weaknesses:**

1. The core contribution lies more in system design. The proposed ADG and FAM modules are essentially heuristic procedures (revision by analogy, empirical counting of failed/successful actions) that combine ideas from prior memory-based and failure-driven learning frameworks (e.g., Reflexion, DECKARD, ADAM).
While the integration is clean, the novelty at the algorithmic level is modest.

2. The evaluation is entirely confined to Minecraft-like environments, where planning graphs and item dependencies are discrete and well-defined. It remains unclear whether the proposed mechanisms generalize to less structured or continuous domains (e.g., physical robot control, household tasks).

3. Although ablation studies are included, some aspects (e.g., sensitivity to hyperparameters like $c_0$, $\alpha_i$, $x_0$) are not deeply explored. It is also unclear how well the algorithm scales when the number of items or actions grows substantially.

4. While the paper argues for learning dependencies purely from interaction, it does not explain why using external structured sources such as Minecraft Wiki is undesirable or infeasible. Since such knowledge bases can already provide accurate dependency graphs, the motivation for building and correcting knowledge from scratch appears weak or artificially constrained. A discussion comparing learning from experience vs. retrieving from knowledge bases would strengthen the motivation.

**Questions:**

see weakness

---

> ### Author Response · Authors · 2025-11-16
>
> ### Contribution and novelty of XENON
>
> We appreciate the reviewer for providing comments that help us clarify the contribution and novelty of XENON, and we provide this clarification below.
>
> * **W1. (contribution and novelty)** We first acknowledge that ADG and FAM are indeed influenced by prior memory-based and failure-driven methods such as DECKARD and Reflexion. However, our core contribution is to propose experience-based algorithmic knowledge correction as an alternative idea to LLM self-correction, for correcting LLM's flawed knowledge. Existing memory-based/failure-driven approaches that use LLMs typically rely on the LLM itself to revise its wrong knowledge [1, 2], and recent work reports that such self-correction can be brittle [3]. We also clarify that ADAM and DECKARD do not correct the LLM’s flawed knowledge (see L105 in our revised manuscript).
>
> While the realization of this idea in ADG, RevisionByAnalogy, could be viewed as a heuristic, its novelty lies in applying the well-established principle of "leveraging similar successful experiences [4, 5]" to correct a flawed LLM-generated knowledge graph *without any additional LLM queries*. To the best of our knowledge, applying this principle to correct flawed LLM-generated knowledge graphs without further LLM calls has been largely unexplored in prior LLM-based agent frameworks. Our ablation experiments show that removing RevisionByAnalogy noticeably degrades performance (Table 5), highlighting its importance for correcting LLM-generated knowledge and enabling robust knowledge learning.
>
> [1] Shinn et al., Reflexion: Language Agents with Verbal Reinforcement Learning, NeurIPS 2023.
>
> [2] Chen et al., "AutoManual: Constructing Instruction Manuals by LLM Agents via Interactive Environmental Learning", NeurIPS 2024
>
> [3] Stechly et al., "On the Self-Verification Limitations of Large Language Models on Reasoning and Planning Tasks", ICLR 2025.
>
> [4] Zhao et al., "ExpeL: LLM Agents Are Experiential Learners", AAAI 2024.
>
> [5] Wang et al., "Agent Workflow Memory", ICML 2024.

---

> ### Author Response · Authors · 2025-11-16
>
> ### Additional experiments
>
> We thank the reviewer for suggesting these additional experiments. In the revised manuscript, we (i) added experiments on a non-Minecraft domain and (ii) conducted a hyperparameter sensitivity study. We are also running further experiments on (a) XENON’s scalability to larger item/action spaces and (b) a scenario where the initial graph is initialized from an external source but the environment’s ground-truth dependencies are perturbed, to show the necessity of knowledge correction.
>
> * **W2. (generalizability of XENON)** We added experiments on a text-based household task planning environment, Microsoft TextWorld Cooking (distinct from our Minecraft-based MC-TextWorld), where the agent must discover valid actions and preconditions for a plan to complete an episode. The results show that (1) XENON generalizes to a domain where the agent must discover valid symbolic actions and preconditions (albeit still in a discrete, symbolic setting rather than a continuous domain), and (2) XENON is more robust than both DECKARD (which does not correct flawed LLM priors) and LLM self-correction, achieving an average success rate of 1.00, while DECKARD and LLM self-correction achieve success rates of 0.09 and 0.75, respectively. Details are in Appendix A. Moreover, we wish to clarify that Minecraft, our primary evaluation domain, captures the core and practical challenges of planning problems. It features (i) long-horizon, knowledge-intensive goals, (ii) valid actions and preconditions for each goal [1], and additionally, (iii) a pre-trained low-level controller that lets us focus on the planner’s knowledge learning and correction. Extending XENON to more diverse and less structured domains is an important direction for future work.
>
> * **W3. (hyperparameter sensitivity)** We added a section presenting an ablation study on hyperparameter sensitivity (Sec. 5.6). In MC-TextWorld and MineRL, we vary each hyperparameter and report the performance of dependency and action learning (EGA) (Figures 7 and 8). The results show that (i) XENON remains stable over a wide range of hyperparameter values---maintaining near-perfect EGA in MC-TextWorld and EGA near or above 0.5 in MineRL---and (ii) even with suboptimal hyperparameters in MineRL, XENON still consistently outperforms all baselines, whose EGA plateaus around or below 0.4.
>
> * **W3. (scalability to larger item/action spaces)** We are running additional experiments in MC-TextWorld where we increase the number of goal items and available actions and measure how the convergence speed of dependency and action learning changes. We will report these results as soon as the experiments are completed.
>
> * **W4. (discussion about external knowledge source)** While we agree that external knowledge sources (e.g., human-curated wikis) can be useful when available, such sources can be flawed or outdated. Therefore, even when an external source is used to initialize the planner’s knowledge, we believe that a method for correcting that knowledge from interaction remains essential, and in domains without such sources the ability to robustly learn knowledge from interaction becomes even more important. To support this, we are running additional experiments where the dependency graph is initialized from an oracle graph from wiki data while the environment’s ground-truth dependencies are perturbed, and we evaluate whether XENON and the baselines can adapt and learn the perturbed ground-truth graph.
>
>
> [1] Kambhampati et al., "LLMs Can't Plan, But Can Help Planning in LLM-Modulo Frameworks", ICML 2024

---

> ### Author Response · Authors · 2025-11-21
>
> ### Experiments on External Knowledge Sources and XENON's Scalability
>
> We thank the reviewer for the helpful suggestions, and added two sets of experiments accordingly. (i) Experiment which uses an external source to initialize the agent’s knowledge shows that, even in this setting, *experience-based knowledge correction remains crucial*, and that *XENON remains applicable and robust* in this scenario compared to other agents that either never modify the given knowledge or rely on unreliable LLM self-correction. (ii) Scalability experiments show that *XENON's dependency and action learning perform well as the numbers of goals and actions increase*, while the number of environment steps needed for convergence grows moderately.
>
> **W4 (external knowledge source to initialize agent's knowledge)**
>
> To support our claim that experience-based knowledge correction remains essential even when an external source is used to initialize an agent's knowledge---since such sources can be flawed or outdated---we add an experiment where the agent’s dependency graph is initialized from an oracle graph while the environment’s ground-truth dependency graph is perturbed. In this setting, obtaining many goal items requires the agent to not blindly trust the given knowledge and instead revise it based on experience.
>
> We evaluate **XENON**, LLM self-correction (**SC**), **DECKARD**, **ADAM**, and **RAND** as in our main experiments and measure the ratio of the 67 goal items obtained *within a single episode*, as shown in the table below:
>
> | Metric                       | XENON | SC   | DECKARD | ADAM | RAND |
> |------------------------------|------|-----|--------|-----|-----|
> | Ratio of obtained goal items | 0.97  | 0.12 | 0.12    | 0.07 | 0.12 |
>
> By continually revising its dependency knowledge, XENON achieves a much higher ratio of goal items obtained in an episode than all baselines. The baselines perform especially poorly because goal items are coupled through dependencies: failing to obtain one goal item makes it impossible to obtain all other goal items that depend on it. The detailed experimental setup and the corresponding learning curves are provided in Appendix K.8 and Figure 23.
>
> **W3 (Scalability of XENON to more goals and actions)**
>
> XENON scales to larger goal and action spaces, which we study using additional dependency and action learning experiments that increase the number of goal items from 67 to 100 and 120, and the size of the action space from 3 to 15, 30, and 45. When varying the number of goals, we keep the original action space of size 3 fixed; when varying the size of the action space, we keep the 67 original goal items fixed.
>
> In all settings, XENON consistently achieves near-perfect EGA, where EGA is the fraction of goal items for which XENON has learned the correct dependency out of all goal items, although the number of environment steps required for convergence naturally differs. The tables below report how many environment steps XENON needs for EGA to converge in MC-TextWorld.
>
> | # goals | Steps to converge |
> |--------:|-------------------|
> | 67      | ~1,400            |
> | 100     | ~2,100            |
> | 120     | ~2,600            |
>
> | # actions | Steps to converge |
> |----------:|-------------------|
> | 3         | ~1,400            |
> | 15        | ~4,000            |
> | 30        | ~7,000            |
> | 45        | ~10,000           |
>
> Particularly, as the action space grows, the number of environment steps to convergence increases noticeably. This is expected because XENON only revises an item's dependency after all available actions for that item have been classified as empirically invalid (i.e., the agent has failed at least $x_0$ times to obtain the item via each action). We believe this convergence speed can be improved with minimal changes, such as lowering the hyperparameter $x_0$, the failure count threshold for classifying an action as empirically invalid.
>
> The detailed experimental setup and the corresponding learning curves are provided in Appendix K.9 and Figure 24.

---

> > ### Comment · Reviewer_S6YL · 2025-11-26
> >
> > Thanks for the clarifications. My main concerns remain mostly about generality.
> >
> > Given that the primary submission area is applications to robotics, autonomy, and planning, XENON’s mechanisms appear closely tailored to Minecraft’s discrete item graph and small action set, so it is still unclear how well the approach would transfer to more complex or continuous real-world settings.
> >
> > Also, evaluating the method in a setup that intentionally avoids available knowledge bases seems somewhat artificial; a more realistic comparison would let all methods use the same external knowledge and examine how well they correct or update it.
> >
> > Finally, the added text-based environment is appreciated, but it is still quite simple compared to robotics or autonomy tasks, so the broader applicability is not yet fully demonstrated.
> >
> > I also remain somewhat reserved about the level of novelty.

---

> > > ### Author Response · Authors · 2025-11-26
> > >
> > > We thank the reviewer for raising the concern regarding generality in complex, continuous settings. However, we wish to clarify that the Minecraft environments (MineRL and Mineflayer) we study are continuous 3D environments, and Minecraft itself is widely regarded as a challenging simulated benchmark for embodied agents [1, 2]. We view XENON as a crucial stepping stone toward practical LLM-based agents in more complex environments for two reasons: (i) in this challenging setting, we empirically find that existing LLM-based methods, when using a small 7B planner LLM, fail to reliably correct their knowledge and struggle to solve long-horizon goals, and (ii) our experience-based knowledge correction enables an agent with a small 7B LLM to autonomously acquire knowledge and significantly outperform agents that rely on much larger proprietary LLMs on long-horizon goal benchmarks. We also note that recent robotics work on continuous environments widely adopts modular architectures where an LLM planner takes discrete symbolic inputs and makes plans by selecting high-level actions from a fixed action set [3, 4].
> > >
> > > On the point about avoiding external knowledge bases, we wish to clarify that we have already reported experimental results with exactly the setup you describe in our previous response. In the experiment described under “W4 (external knowledge source to initialize agent’s knowledge),” the dependency graph for all methods is initialized from the same graph derived from wiki data, and then the environment’s ground-truth dependencies are perturbed. LLM self-correction (SC) is allowed to revise its own knowledge when plans fail, but it fails to revise its knowledge to match the perturbed environment and therefore obtains far fewer goal items than XENON.
> > >
> > > Finally, when solving long-horizon goals with an oracle dependency graph, XENON’s efficiency does not degrade as the action space grows. As we show in our response to Reviewer VSc5, even when the action space increases from 3 to 45 actions in MC-TextWorld, XENON maintains a success rate of 1.0 and the number of environment steps per successful episode stays around 50, so the number of LLM calls per episode also remains nearly constant. Details are provided in Appendix K.10 and Figure 25.
> > >
> > >
> > > [1] Zhou et al., “MineDreamer: Learning to Follow Instructions via Chain-of-Imagination for Simulated-World Control”. IROS 2025.
> > >
> > > [2] Qin et al., "MP5: A Multi-modal Open-ended Embodied System in Minecraft via Active Perception". CVPR 2024.
> > >
> > > [3] Ravichandran et al., "SPINE: Online Semantic Planning for Missions with Incomplete Natural Language Specifications in Unstructured Environments". ICRA 2025.
> > >
> > > [4] Ao et al., "LLM-as-BT-Planner: Leveraging LLMs for Behavior Tree Generation in Robot Task Planning". ICRA 2025

---

### Official Review · Reviewer_DZb7 · 2025-10-24

**Soundness:** 2
**Presentation:** 3
**Contribution:** 3
**Rating:** 6
**Confidence:** 5

**Summary:**

The paper proposes XENON, an LLM-based agent that algorithmically corrects its external planning knowledge from interaction experience, aiming to overcome flawed LLM priors and sparse feedback in long-horizon Minecraft tasks. Two key components drive the method: (1) an Adaptive Dependency Graph (ADG) that initializes a dependency graph using an LLM and then revises it from successful trajectories via a procedure called RevisionByAnalogy; and (2) a Failure-aware Action Memory (FAM) that maintains empirical success/failure counts per (item, action) to prune invalid actions, explore under-tried ones, and trigger ADG revisions when all actions for an item are deemed invalid. The agent is evaluated across MineRL and Mineflayer. Experimental results show that XENON outperforms other baselines. With a 7B open-weight planner (Qwen2.5-VL-7B), it attains competitive or superior SR compared with agents that rely on larger proprietary models.

**Strengths:**

1. The paper tackles a real failure mode: LLM priors over item dependencies and actions are brittle and hard to self-correct with only binary feedback. Treating the LLM as a planner while moving correction into algorithmic external memory is a crisp, testable design choice.

2. Using Qwen2.5-VL-7B, XENON outperforms or rivals methods using GPT-4/4V on several long-horizon task groups, particularly when oracle dependencies are provided. It still performs strongly with learned dependencies on challenging tasks.

3. This paper provides a detailed account of the motivation and methodology, along with comprehensive implementation details and experimental setup, making it easy to follow.

**Weaknesses:**

1. The proposed method is simple and intuitive, akin to prompting an LLM to correct memory and graphs, lacking technical innovation.

2. The procedure references “similar, successfully obtained items” but the similarity function, features, and retrieval specifics are not fully spelled out in the main text (embedding choice, distance metric, negatives, and sensitivity). This matters because ADG’s replacement set can systematically bias learning if similarity is noisy.

3. The design hinges on $c_0$, $x_0$, $a_i$, $a_s$. While intuitively motivated (hallucination handling, controller tolerance), the paper does not present comprehensive ablations on these hyperparameters or learning stability across seeds, leaving robustness to tuning somewhat unclear.

4. The approach is tested only in Minecraft variants; claims about “practical embodied agents” would be stronger with at least one additional domain (e.g., household manipulation or web-based embodied tasks) or a clearer argument for portability.

**Questions:**

Table 3 shows that some baselines, such as Optimus-1, outperform XENON on certain simple task groups, while XENON significantly outperforms the baseline on difficult task groups like Diamond. What causes this phenomenon? Are the experimental settings consistent?

---

> ### Author Response · Authors · 2025-11-16
>
> ### Clarifications of novelty and similarity notion
>
> We thank the reviewer for constructive comments that help us clarify our work. In summary, we (i) clarify the technical novelty of XENON as an algorithmic knowledge correction method for LLM-generated knowledge without further LLM prompting, and (ii) specify the similarity notion used in RevisionByAnalogy, which is now explicitly described in the revised manuscript.
>
> * **W1. (technical novelty)** We wish to clarify that we do *not* prompt an LLM to correct knowledge (memory or graphs). Specifically, after we initialize the Adaptive Dependency Graph (ADG) using an LLM, XENON does not query the LLM to correct it; all subsequent corrections are performed algorithmically from interaction experience. We also clarify that our core contribution is to propose experience-based algorithmic knowledge correction as an alternative idea to LLM self-correction for repairing flawed LLM-generated knowledge.
>
> From a technical perspective, our novelty lies in applying the well-established principle of "leveraging similar successful experiences [1, 2]" to correct a flawed LLM-generated knowledge graph *without any additional LLM queries*. To the best of our knowledge, applying this principle to correct flawed LLM-generated knowledge graphs without further LLM calls has been largely unexplored in prior LLM-based agent frameworks. Our ablation experiments (Table 5) show that removing RevisionByAnalogy noticeably degrades performance, highlighting its importance, while LLM self-correction fails to effectively correct flawed knowledge in our setting.
>
> [1] Zhao et al., "ExpeL: LLM Agents Are Experiential Learners", AAAI 2024.
>
> [2] Wang et al., "Agent Workflow Memory", ICML 2024.
>
> * **W2. (similarity notion)** In the revised manuscript (L235), we explicitly described the similarity function used in RevisionByAnalogy: we compute similarity between items using Sentence-BERT embeddings of their names with cosine similarity, and use the top $K=3$ most similar successfully obtained items as references when proposing new dependencies. We also note that even if ADG occasionally selects somewhat dissimilar items as references, XENON still learns dependencies robustly as long as the controller is competent. For example, XENON successfully learns the dependencies for the Redstone goal group, where item names are dissimilar (e.g., dropper, note block). Despite this challenge, XENON, when combined with competent controllers in Mineflayer and MC-TextWorld, still learns the dependencies for the Redstone goals, and we provide a more detailed analysis in Appendix K.3.

---

> ### Author Response · Authors · 2025-11-16
>
> ### Additional experiments: hyperparameter sensitivity and generality
>
> We thank the reviewer for suggesting important additional experiments; in summary, we report new experiments on XENON’s hyperparameter sensitivity and add experiments on a domain other than Minecraft.
>
> * **W3 (hyperparameter sensitivity)** We added a section presenting an ablation study on hyperparameter sensitivity (Sec. 5.6). In MC-TextWorld and MineRL, we vary each hyperparameter and report the performance of dependency and action learning (EGA) (Figures 7 and 8). The results show that (i) XENON remains stable over a wide range of hyperparameter values---maintaining near-perfect EGA in MC-TextWorld and EGA near or above 0.5 in MineRL---and (ii) even with suboptimal hyperparameters in MineRL, XENON still consistently outperforms all baselines, whose EGA plateaus around or below 0.4.
>
> * **W4. (generality of XENON)** We added experiments on a text-based household task planning environment, Microsoft TextWorld Cooking (distinct from our Minecraft-based MC-TextWorld), where the agent must discover valid actions and preconditions for a plan to complete an episode. The results show that (1) XENON generalizes to a domain where the agent must discover valid symbolic actions and preconditions, and (2) XENON is more robust than both DECKARD (which does not correct flawed LLM priors) and LLM self-correction, achieving an average success rate of 1.00, while DECKARD and LLM self-correction achieve success rates of 0.09 and 0.75, respectively. Details are in Appendix A.
>
> We also wish to clarify that, by practicality for embodied agents, we mean the ability to plan and succeed tasks robustly even with relatively small LLMs, which we view as crucial because hardware constraints in embodied settings often make it difficult to deploy very large models.
>
>
> ### Clarification on experimental settings and baseline comparisons
>
> * **Q1. (experimental setting)** We thank the reviewer for the detailed question regarding our experimental setting. Our experimental settings are consistent: we use the official Optimus-1 implementation without any changes to the environment setup or evaluation. Below, we explain why some baselines do slightly better than XENON on a simple group and why XENON outperforms them on other groups.
>
> The only group where any baseline outperforms XENON is the Wood group in Table 3, and this difference mainly comes from the underlying planner/controller models. In the Optimus-1 paper, when the planner is reduced to a 7B model (DeepSeek-VL 7B), i.e., the same model size used in XENON, the reported Wood success rate drops below 0.93, which is lower than XENON’s 0.95. For Optimus-2, the higher Wood group success rate is due to a more capable controller than STEVE-1.
>
> XENON outperforms the baselines in other groups for two reasons (as discussed in L401): (i) FAM provides action correction for each goal, and (ii) XENON reduces the burden on the LLM planner by shortening input prompts and outputs (i.e., predicting one action per subgoal item), instead of asking the LLM to produce long plans from long prompts. Our error analysis of Optimus-1$^\dagger$ (Appendix K.5) supports this, showing that most failures come from choosing invalid actions and omitting required subgoal items when generating long plans from long prompts.
>
> Additionally, we note that our main MineRL experiments use the same modified MineRL configuration as the official Optimus-1 codebase: key resource items (iron, gold, diamond) are easier to obtain---a common modification in Minecraft planning works (see Appendix J.2.5) to better isolate the planning problem from the controller’s ability to execute plans. In Appendix K.4, we additionally evaluate DEPS, Optimus-1, and XENON in the standard MineRL environment, where such resource items are difficult to obtain, and observe that XENON still consistently achieves higher success rates than the baselines, although all methods’ performance is heavily limited by the controller’s capacity.

---

> > ### Comment · Reviewer_DZb7 · 2025-11-17
> >
> > Thank you for the explanation. I still have two concerns:
> >
> > 1. The author needs to elaborate in detail on the composition of the entire agent framework and the input/output streams for inference within the main text.
> >
> > 2. For a long-horizon task, how many times does the LLM need to be queried to obtain the sequence of sub-goals (or is each sub-goal queried once)? How many times can each sub-goal be corrected by FAM (or is it corrected until the sub-goal is finalized)? Is each sub-goal attempted only once, or should multiple attempts be made to interact with the environment and correct through FAM?

---

> ### Author Response · Authors · 2025-11-21
>
> **W1. (input/output specification in the main text)** We thank the reviewer for this helpful suggestion to improve the clarity of our paper. In the revised manuscript, we made the inference-time input/output streams explicit both at the overall agent level and at the component level, as summarized below:
>
> - (L166) In the opening paragraph of the Methods section, we now explicitly describe the input and output streams of each module in XENON's learning pipeline.
> - (L206) We explicitly describe the input and output of ADG’s RevisionByAnalogy procedure.
> - (L268) We clarify the input and output of our action-selection procedure using the planner LLM and FAM.
> - (L285) We specify the input and output of the context-aware reprompting (CRe).
>
>
> **W2. (LLM call frequency and FAM's action correction)** We thank the reviewer for this clarifying question. For each item in a subgoal, XENON keeps querying an LLM until XENON finds a valid action that successfully obtains that item, so the LLM may be called multiple times for some items. Therefore, with an empty failure-aware action memory (FAM) (i.e. no successful actions stored), the number of LLM calls for a long-horizon task is approximately, `# of LLM calls ≈ (# of distinct items to obtain) + (# of times the LLM selects an invalid action for those items).` For example, when the agent needs to obtain 13 distinct items and starts from an empty memory, XENON typically makes about 15–17 LLM calls per episode in our MC-TextWorld experiments.
>
> Here, *XENON substantially reduces the total number of LLM calls* because (i) FAM excludes actions that have repeatedly failed to obtain a given item from the set of candidate actions that the LLM can choose from, and (ii) once a valid action has been found, the valid action is stored in FAM, and future subgoals that require the same item are constructed without querying the LLM, simply reusing the stored valid action in FAM.
>
> In our experiments, this reduction in LLM calls and reliable discovery of valid actions was only possible when the LLM’s flawed action knowledge was explicitly corrected, as done by XENON's FAM; in contrast, LLM self-correction (SC) cannot fix its flawed action knowledge. SC fails to find valid actions for some items (e.g., "redstone") and thus fails to solve the corresponding long-horizon goals within the fixed episode horizon.

---

> > ### Comment · Reviewer_DZb7 · 2025-11-24
> >
> > Thank you for the clarification. However, if the inference phase requires multiple queries to the LLM or the use of FAM to refine each sub-goal, then the experimental comparison is not fair. Baselines in Table 3 such as Jarvis-1 only need to query the LLM once per sub-goal, while Optimus-1 requires only a single query to obtain the complete sub-goal sequence. Allowing XENON to refine its sub-goal sequence through multiple queries gives it an unfair advantage over the baselines. Alternatively, the inference time per episode should be taken into account when comparing the methods.

---

> ### Author Response · Authors · 2025-11-25
>
> We thank the reviewer for raising this concern about fairness and computational efficiency, and we fully agree that computational efficiency is crucial in practice. First, we respectfully clarify that the baselines in Table 3 also call the planner LLM multiple times within a single episode to re-plan when their initial plan fails, even though the initial plan itself is generated in a single LLM call. For example, the JARVIS-1 paper reports that it typically requires about 2–3 rounds of re-planning, and Optimus-1 similarly calls the LLM multiple times for re-planning on failures.
>
> To examine computational efficiency, we compare the LLM usage of XENON and Optimus-1 in MC-TextWorld through an additional experiment. In this controlled setting, *when both agents use the same 7B LLM planner, we observe that XENON uses fewer LLM calls and noticeably fewer input/output tokens, resulting in a shorter LLM inference wall-clock time per episode than Optimus-1*. In this experiment, both agents are given the same oracle dependency graph; at the beginning of each episode, their memories are initialized with three human-written plans ("craft iron_sword", "craft golden_sword", "mine diamond"), a goal is given as one of the six Redstone-group goals, and the episode horizon is set to 100 steps. We run 10 episodes per goal (60 episodes in total) and report averages over all episodes:
>
>
> | Method    | Success rate | # of LLM calls | LLM input tokens | LLM output tokens | LLM inference wall-clock time |
> |-----------|--------------|-------------|------------------|-------------------|-------------------------------|
> | XENON     | 1.00         | 4.2         | 1,191            | 83                | 2.6 s                         |
> | Optimus-1 | 0.67         | 4.9         | 2,722            | 1,414             | 38.6 s                        |
>
>
> The large number of LLM input/output tokens for Optimus-1 comes from two factors: (i) at both the initial planning and every re-planning, Optimus-1 feeds the LLM a long prompt and asks for a long plan as output, and (ii) the 7B LLM struggles to handle such long inputs/outputs and often produces very long but meaningless plans that repeatedly restate the same subgoal.
>
> Finally, we note that Optimus-1's long-horizon goal success rates in MineRL (Table 3) are much lower than in this MC-TextWorld experiment because planning errors are far more harmful in MineRL, where an imperfect controller is used. Even if re-planning eventually produces a correct plan, the agent may already be in a state from which the controller cannot reliably follow the revised plan (e.g., the new plan requires the agent to return to the ground even though it is already deep underground), whereas in MC-TextWorld a perfect controller can reliably execute the revised plan.

---

### Official Review · Reviewer_JH8W · 2025-10-31

**Soundness:** 3
**Presentation:** 3
**Contribution:** 2
**Rating:** 4
**Confidence:** 3

**Summary:**

This paper proposes XENON, a method for learning Minecraft recipes and using this learned knowledge to solve Minecraft problems. XENON uses an LLM to generate an initial set of recipes (the graph). Then, it refines it by trying to use it and observing if it works or not. If it does not work, XENON tries to modify the recipe. After sufficient amount of tries, XENON will give up and state that the item is an hallucination. The author evaluate this on a standard set of Minecraft tasks from the MineRL benchmark and a text version of Minecraft.

**Strengths:**

1.	Learning the recipes by interacting with the environment makes sense and I appreciate its integration with the LLM agent.
2.	The results show nice improvements over the ablation and baseline.
3.	The solution is designed to handle execution delays

**Weaknesses:**

1.	The paper is not sufficiently self-contained. In particular, there authors assume the reader knows how DECKARD works (e.g.,in line 172). Note that DECKARD was only published in Arxiv and it not well known.
2.	I am not convinced about the generality of the approach because it relies on hyper parameters that I guess are tuned for this domain.
3.	Some details are not clear, and there are quite a few design choices that seem arbitrary to me. See below in the list of questions.

**Questions:**

1.	Lines 127-128: It seems your graph does not support cases where crafting N items of one type create M>1 items of another type, since you only have q for the quantity of the resource, but no parameter for the quantity of the resulting items. For example, consider the case where CRAFT on two logs create four sticks.
2.	Line 138: When the controller executions these subgoals” – how does the controller do this?
3.	Were the parameters of XENON tuned differently for each test goal?
4.	What is a “context-aware reprompting” technique mentioned in line 268?

Line 22: What makes an LLM be “open-weight”?
Lines 84-90: The first and second contribution are actually only one contribution. What is the difference between them? (one proposes XENON, the other explains how XENON works…)
Line 104: What is a “… robust action correlation”?
Line 114: What is a “parametric knowledge”?
The EGA metric: isn’t it just the mean over the metric you mentioned in (1)? Why do we need a new name for this metric? (line 288)

---

> ### Author Response · Authors · 2025-11-16
>
> ### Hyperparameters and generality
>
> We thank the reviewer for comments about hyperparameters and generality, which helped us strengthen the empirical support for our work. In summary, we added an ablation study on hyperparameter sensitivity and additional experiments in a non-Minecraft domain.
>
> * **W2 & Q3. (hyperparameter)** We wish to clarify that all hyperparameters are kept the same across all experiments. We added a section presenting an ablation study on hyperparameter sensitivity (Sec. 5.6). In MC-TextWorld and MineRL, we vary each hyperparameter and report the performance of dependency and action learning (EGA) (Figures 7 and 8). The results show that (i) XENON remains stable over a wide range of hyperparameter values---maintaining near-perfect EGA in MC-TextWorld and EGA near or above 0.5 in MineRL---and (ii) even with suboptimal hyperparameters in MineRL, XENON still consistently outperforms all baselines, whose EGA plateaus around or below 0.4.
>
> * **W2. (generality)** We added experiments on a text-based household task planning environment, Microsoft TextWorld Cooking (distinct from our Minecraft-based MC-TextWorld), where the agent must discover valid actions and preconditions for a plan to succeed an episode. The results show that (1) XENON generalizes to a domain where the agent must discover valid symbolic actions and preconditions, and (2) XENON is more robust than both DECKARD (which does not correct flawed LLM priors) and LLM self-correction, achieving an average success rate of 1.00, while DECKARD and LLM self-correction achieve success rates of 0.09 and 0.75, respectively. Details are in Appendix A.
>
> ### Comments on Clarity and Presentation
>
> We deeply appreciate the reviewer’s detailed comments on clarity and presentation, which helped us improve the self-containment of the manuscript. We revised the paper accordingly. Below we summarize how we addressed each point.
>
> * **W1. (DECKARD description)** We provided a more detailed description of DECKARD and clarified how it differs from XENON in both the related work (L106) and the dependency graph initialization section (L176).
> * **Q1. (multiple resulting item quantities in the dependency graph)** We added a clarifying footnote at L161 stating that our actual implementation stores the resulting item quantity on each edge, even though we omit this term from the notation for simplicity.
> * **Q2. (how the controller executes subgoals)** We clarified at L137 that the controller receives language subgoals as input and executes low-level actions in the environment to achieve them.
> * **Q4. (definition of context-aware reprompting)** We clarified the notion of context-aware reprompting at L283: it is a technique that replaces the controller’s language subgoal for item acquisition with a temporary subgoal to escape a stalled state.
> * **Q5. (meaning of "open-weight")** We use the term 'open-weight' to refer to models whose parameters (weights) are publicly released, as opposed to proprietary models that are accessed via paid APIs.
> * **Q6. (overlap between the first and second contributions)** We merged the original first and second contributions into a single contribution at L100 to avoid confusion, while explicitly separating our main idea (experience-based algorithmic knowledge correction instead of LLM self-correction) from its concrete realization via ADG and FAM.
> * **Q7. (meaning of "robust action correction")** We revised L105 to clarify this phrase and avoid confusion.
> * **Q8. (meaning of "parametric knowledge")** At its first occurrence (L44), we now give an inline definition: parametric knowledge refers to knowledge implicitly stored in the LLM’s parameters, in contrast to explicit external knowledge memory (our dependency graph and action memory).
> * **Q9. (EGA metric)** At L306, we clarified that EGA is not a new metric but simply a name for the averaged version of the metric introduced in Equation (1), used to provide an intuitive score between 0 and 1 for learning performance.
> * Additionally, to make it easier for readers to navigate the paper, we added pointers from the main text to the appendix: at L245 to the pseudo-code of ADG’s RevisionByAnalogy and at L349 to the appendix section describing the baselines.

---

> > ### Comment · Reviewer_JH8W · 2025-11-19
> > **Thanks!**
> >
> > The authors' response and edits to the paper are helpful and increate the quality of the paper. Thanks!

---

> > > ### Author Response · Authors · 2025-11-21
> > >
> > > As a brief update, to further improve clarity and presentation, we revised the manuscript to make the input/output streams more explicit, as summarized below:
> > >
> > > - (L166) In the opening paragraph of the Methods section, we now explicitly describe the input and output streams of each module in XENON's learning pipeline.
> > > - (L206) We explicitly describe the input and output of ADG’s RevisionByAnalogy procedure.
> > > - (L268) We clarify the input and output of our action-selection procedure using the planner LLM and FAM.
> > > - (L285) We specify the input and output of the context-aware reprompting (CRe).
> > >
> > > We hope these changes further improve the clarity and presentation of the paper, and we would be very happy to receive any additional comments or suggestions you may have.

---

> > > > ### Author Response · Authors · 2025-11-29
> > > > **Dear Area Chair,**
> > > >
> > > > Thank you very much for overseeing the review process of our paper.
> > > >
> > > > We would like to note that on 26 Nov 2025 at 16:00 EST, **Reviewer JH8W increased their rating from 4 to 8** and stated in the updated review: *"POST-REBUTTAL I find the clarifications made by the reviewer helpful, and increase my score accordingly."* (we believe "the reviewer" here is a typo for "the authors").
> > > >
> > > > We simply wanted to bring this to your attention so that you are aware that this reviewer's post-discussion evaluation was more positive than what is currently reflected in the rolled-back rating and review text.
> > > >
> > > > Thank you again for your time and efforts.

---

### Official Review · Reviewer_VSc5 · 2025-10-31

**Soundness:** 2
**Presentation:** 2
**Contribution:** 2
**Rating:** 4
**Confidence:** 4

**Summary:**

The paper introduces XENON, an agent that provides a training-free solution to revise knowledge algorithmically from experience without changing the parameters of the LLM. This addresses hallucinations caused by flawed priors inherent in LLMs. Specifically, the paper models knowledge as a directed acyclic graph, where nodes represent Minecraft items and directed edges represent dependence and actions. The model initially uses an LLM to distill and initialize an imperfect Dependency Graph, which is then adaptively revised using the RevisionByAnalogy technique. If a dependency fails repeatedly during rollouts, it is deleted, while for less-tried items, a similar, previously obtained item is sampled to replace the original dependency. Action selection is optimized using Failure-aware Action Memory (FAM) through an elimination method.
The agent is validated using 7B-qwen2.5-vl in three different environments—MineRL, Mineflayer, and MC-TextWorld. The results show that XENON is uniquely robust to LLM hallucinations and adapts to different control policies, achieving state-of-the-art performance in the same domain.

**Strengths:**

- The paper proposes a novel algorithmic approach for planning that does not require retraining LLMs, offering an efficient solution to improve LLMs’ knowledge representation from experience.
- The ablation study clearly demonstrates that Adaptive Dependency Graph (ADG) and FAM are highly robust to initial errors. They effectively recover the correct dependency graph despite the presence of flawed priors. Moreover, no additional LLMs are required to generate these priors, which greatly enhances the efficiency of the ablation. The separate analysis of ADG and FAM’s impacts aids in understanding their individual contributions. This constitutes a well-designed experiment to validate the framework.
- The visual quality of the figures is commendable. The graphical representations are intuitive and help readers easily grasp the experimental setup and results

**Weaknesses:**

1. The paper states (Line 126) that knowledge is modeled as a directed acyclic graph. However, it is unclear how the system detects and resolves cyclic structures that may arise during initialization ( e.g. when a wooden axe is required to obtain logs, logs produce planks, and planks are used to craft the axe). I suggest clarifying whether cycles are possible in practice and, if so, how they are algorithmically identified and dismantled.
2. In Line 212, the method refers to selecting required items for less-tried nodes based on similarity. However, It seems that the notion of “similarity” is not formally defined in the current draft.
3. The ADG is a relatively complex algorithm, and presenting it only with diagrams results in high information density. It is recommended to include pseudocode for better clarity and reproducibility.
4. The paper discusses both the control policy and the XENON framework in detail; however, it lacks experimental analysis regarding the role of the LLM itself. If the initial dependency graph were directly provided without using an LLM, and actions were sampled using MCTS rather than LLM-based selection, would the agent still be able to complete the planning task? What is the essential contribution of the LLM within XENON? I believe an ablation study in this regard would be valuable.

**Questions:**

1. In the MineRL environment, approximately how many episodes are required for the dependency-action graph to converge? Additionally, while the current experiments focus on 67 goals with a limited number of items, how does the method scale when the item space becomes significantly larger? A brief comment on scalability or computational complexity would be valuable for understanding the broader applicability of XENON.

---

> ### Author Response · Authors · 2025-11-16
>
> ### Clarifications of our manuscript
>
> We thank the reviewer for these helpful comments, which improved the clarity and overall readability of our work. We updated our manuscript to reflect these suggestions and address each comment in turn below.
>
> *  **W1. (cycles in the dependency graph)** We added in the revised manuscript (L196 and Appendix E.2) how XENON handles potential cycles during adaptive dependency graph (ADG) initialization. While cycles can in principle arise during ADG initialization, XENON prevents them algorithmically and maintains the ADG as a directed acyclic graph: whenever we add LLM-predicted incoming edges for a node, we first check whether inserting these edges would create a cycle; if so, we discard those LLM-predicted edges and instead assign an empty incoming edge set to that node.
> * **W2. (similarity definition)** In the revised manuscript (L239), we added the similarity notion used in RevisionByAnalogy: we compute similarity between items using Sentence-BERT embeddings of their names with cosine similarity, and use the top $K=3$ most similar successfully obtained items as references when proposing new dependencies.
> * **W3. (pseudocode of ADG)** We added an explicit pointer at L248 directing readers to the pseudocode of ADG’s RevisionByAnalogy in the appendix for better clarity and reproducibility. This pseudocode was already included in the appendix of the original manuscript, but the missing pointer in the main text likely made it hard to discover.
>
>
> ### Role of the LLM in Action Selection and Scalability of XENON
>
> We thank the reviewer for constructive comments that help us clarify our work and for suggesting interesting experiments. In summary, we empirically observe that XENON converges by around 400 episodes in MineRL, and we are running additional experiments that (i) ablate action selection methods and (ii) vary the number of goal items and available actions to measure how convergence speed changes.
>
> * **W4. (role of LLM in action selection)** While we think that, with our Failure-aware Action Memory (FAM), an agent could eventually succeed on goals even without using an LLM for action selection (e.g., by sampling from the action space and updating FAM), we believe that the LLM can accelerate the exploration of valid actions using its prior knowledge and in-context learning. For in-context learning, we augment the LLM’s action-selection prompt with the valid actions for similar, successfully obtained items retrieved from our FAM to speed up the search for a valid action (we added this at L268 in the revised manuscript).
> To support this, we are running additional experiments where we vary the action selection methods to compare the timesteps required to achieve diverse goals, assuming that the initial dependency graph is given as an oracle (i.e., provided without using an LLM). We will report these results as soon as the experiments are completed.
>
> * **Q1. (convergence of XENON in MineRL)** In MineRL, the dependency graph converges by around 400 episodes. Even when extending learning to 600 episodes, XENON learns no additional dependencies. This stems from the limited capacity of our controller (STEVE-1), which often fails to collect sufficient quantities of items.
>
> * **Q1. (scalability to larger goal item spaces)** We are currently running experiments in MC-TextWorld where we increase the number of goal items and measure how the timesteps required for convergence of dependency and action learning (EGA over timesteps) change with the size of the item space. Additionally, we are varying the number of available actions to examine how convergence speed changes as the action space grows. We will report these results as soon as the experiments are completed.

---

> ### Author Response · Authors · 2025-11-21
>
> ### Role of the LLM for action selection and Scalability of XENON
>
> We sincerely thank the reviewer for the constructive suggestions, and have added new experiments accordingly. (1) An ablation on action selection methods shows that an LLM can accelerate the search for valid actions when making subgoals through its prior knowledge, but *only when its flawed knowledge is corrected by our Failure-aware Action Memory (FAM)*, a memory that, for each item, tracks the success/failure history of actions and prevents LLM from selecting actions that have repeatedly failed. Without such correction, LLM-based methods perform worse than action selection methods not using an LLM. (2) A scalability study shows that XENON’s dependency and action learning maintain near-perfect performance as the goal space grows (from 67 to 100 and 120 goals), with only moderate increases in the number of environment steps required for convergence.
>
> **W4 (role of an LLM and non-LLM baselines in action selection for making subgoals).**
>
> We run a new ablation where the agent is given a long-horizon goal and an oracle dependency graph, so it only needs to select one valid action from the action space for each subgoal item in order to solve the long-horizon goal (i.e., successfully obtain the goal item). In this setting, we vary the size of the action space and compare five action selection methods: (i) **Random+FAM**, which randomly samples from actions that have not yet repeatedly failed and reuses past successful actions; (ii) **UCB**; (iii) **LLM** without memory; (iv) LLM self-correction (**SC**); and (v) **XENON**, which combines the LLM with FAM. The table below reports the average success rate and the average number of environment steps per successful episode in MC-TextWorld for the largest action space setting we test (45 actions).
>
>
> | Method | Success rate | Steps to succeed (lower is better) |
> |-----------|-------------|------|
> | Random+FAM | 0.02 | 286.0 |
> | UCB | 0.34 | 277.9 |
> | LLM | 0.00 | – (no success) |
> | SC | 0.00 | – (no success) |
> | XENON | 1.00 | 53.2 |
>
> The detailed experimental setup and full results are provided in Appendix K.10 and Figure 25 of the revised manuscript; here we summarize the main trends of results. Across all action space sizes we test (3, 15, 30, 45), **XENON** consistently achieves a success rate of 1.0 and reaches the goal in around 50 environment steps, regardless of how large the action space is. In contrast, the non-LLM baselines **Random+FAM** and **UCB** perform well when the action space is small, but as the action space grows their success rates drop and they require more steps per successful episode. LLM and SC never succeed in any episode for any action space size.
>
> Finally, regarding the reviewer’s suggestion to use MCTS, in our setting, (i) each subgoal item has a fixed valid action and (ii) the outcome of a tried action is observed easily as a binary signal (successfully obtaining the item or not). This makes the problem a natural fit for an action-level multi-armed bandit rather than a tree-search problem, so we use UCB as a non-LLM baseline instead of MCTS.
>
> **Q1 (scalability in the number of goals).**
>
> XENON scales to larger goal spaces, and we examine this by running additional dependency and action learning experiments while increasing the number of goal items it must learn. We increase the number of goal items from 67 to 100 and 120 by adding new goal items, and report how many environment steps are needed for EGA to converge in MC-TextWorld, where EGA is the fraction of goal items for which XENON has learned the correct dependency out of all goal items:
>
> | # goals | Steps to converge |
> |-------|------|
> | 67 | ~1,400 |
> | 100 | ~2,100 |
> | 120 | ~2,600 |
>
> We report the detailed experimental setup for these experiments and additionally show how XENON's dependency and action learning scale with the size of the action space in Appendix K.9 and Figure 24.

---

> > ### Comment · Reviewer_VSc5 · 2025-11-27
> >
> > Thank you for the author's reply, my main concern has been resolved. I will consider increasing the final score.

---

### Author Response · Authors · 2025-11-24

We sincerely appreciate the reviewers' constructive comments and suggestions, which have helped us further clarify and enrich our work. To better highlight XENON's practicality and robustness even with a smaller planner LLM (4B instead of 7B), we added an experiment showing that XENON can robustly learn dependency and action knowledge. We would be very happy to receive any further comments and continue the discussion.

### Additional experiment: robustness to smaller planner LLMs

We evaluate **XENON** and baselines (LLM self-correction (**SC**), **DECKARD**, **ADAM**) for dependency and action learning in MC-TextWorld under the same setup as our main experiments (no ground-truth rule perturbations, 67 goals, 3 high-level actions), while changing the planner LLM from a 7B model (Qwen2.5-VL-7B) to a 4B model (Phi-4-mini). The table below reports EGA (performance of dependency and action learning) after 3,000 environment steps of learning:


| Method  | 7B LLM  | 4B LLM  |
|---------|------|------|
| XENON   | 0.97 | 0.96 |
| SC      | 0.62 | 0.49 |
| DECKARD | 0.48 | 0.38 |
| ADAM    | 0.68 | 0.53 |

As these results show, XENON maintains near-perfect EGA even with the 4B planner, while methods that rely on LLM self-correction (SC) or simply trust the planner's flawed prior knowledge without correcting it (DECKARD, ADAM) suffer substantial degradation when the LLM size is reduced. More detailed results and plots (including experiments with initial dependency graphs that contain less human-provided knowledge) are provided in Appendix K.11 of the revised manuscript.

---

### Author Response · Authors · 2025-11-29
**Summary of discussion**

We sincerely thank the reviewers for their constructive feedback and the Area Chair for their time and efforts.

We would like to note that as a result of this discussion, Reviewer JH8W **increased their rating from 4 to 8**, and Reviewer VSc5 wrote that "my main concern has been resolved. **I will consider increasing the final score"**. Below, we summarize the strengths recognized by the reviewers and how, after this discussion, we addressed their initial concerns.

## Strengths

**Important and practical problem with a clear idea (JH8W, DZb7, S6YL).**

Reviewers agree that the paper tackles a key failure mode of LLM-based planning: LLMs have flawed prior knowledge and struggle to correct it with prompting alone. They also appreciate the idea of correcting knowledge stored outside the LLM in an external memory, purely from experience and without additional LLM calls, and then using the LLM to plan by consulting this corrected memory as a promising direction for lightweight embodied agents.

**Strong and comprehensive empirical results with a small LLM (VSc5, JH8W, DZb7, S6YL).**

All reviewers acknowledge that the empirical evaluation is extensive and well designed. With a small open-weight 7B LLM, our method robustly corrects the LLM's flawed prior knowledge, learns planning-relevant knowledge, and achieves strong performance on several long-horizon tasks in Minecraft. Reviewer DZb7 further highlights that our method, i.e., XENON, with this 7B LLM matches or outperforms prior agents that rely on much larger models such as GPT-4/4V.

## Initial concerns and how we addressed them

**Clarity and presentation (VSc5, JH8W, DZb7).**

We updated our manuscript to improve self-containment and readability by explicitly describing module inputs/outputs, defining key terms more explicitly, and adding missing details. All revisions are highlighted in blue in the updated manuscript.

**Generalizability beyond Minecraft (JH8W, DZb7, S6YL).**

We added experiments on Microsoft TextWorld Cooking, showing that our method (i) corrects an LLM's flawed planning-relevant knowledge in a non-Minecraft environment and (ii) outperforms LLM self-correction in solving goals in this environment. In response to Reviewer S6YL's generalizability concern about more complex continuous environments, we also clarified that Minecraft, our main environment, is already a complex continuous 3D environment widely regarded as a challenging benchmark to build embodied agents.

**Technical novelty (DZb7, S6YL).**

While Reviewer VSc5 describes our method as “a novel algorithmic approach,” Reviewers DZb7 and S6YL question its technical novelty. In response, we clarified that we propose a novel idea of *experience-based algorithmic knowledge correction* **without any additional LLM calls** as an alternative to LLM self-correction for fixing an LLM's flawed prior knowledge. Our method instantiates this idea by applying the principle of *leveraging similar successful experiences* to repair an LLM-generated knowledge graph without further LLM queries.

**Additional experiments on hyperparameters, scalability, and efficiency.**

In response to the reviewers' suggestions, we added the following experiments:

- **Hyperparameter sensitivity:** XENON is robust to hyperparameter choices in two environments (MC-TextWorld and MineRL).
- **Scalability in goals and actions:** XENON learns planning-relevant knowledge well even when the numbers of goals and actions increase.
- **External knowledge source:** In this setting, XENON and all baselines must learn the environment's planning-relevant knowledge while initializing it from an external source (e.g., a game wiki) that is mismatched with the environment's ground-truth rules. We show that knowledge correction remains necessary even when an external knowledge source is available, and that XENON robustly corrects such misaligned external knowledge while still learning accurate planning-relevant knowledge.
- **Subgoal construction ablation:** Without our algorithmic knowledge correction, LLM-based methods, including LLM self-correction, underperform non-LLM methods such as UCB when solving long-horizon goals.
- **LLM usage efficiency:** XENON is more computationally efficient than a strong baseline in terms of LLM usage (number of calls, tokens, and wall-clock inference time) under the same 7B LLM.
- **Effect of planner LLM size:** XENON maintains near-perfect knowledge-learning performance with a smaller 4B planner LLM, while baselines degrade.


We hope this summary is helpful, and we sincerely thank you again for your time and consideration.

---

### Meta-Review · Area_Chair_2bCz · 2026-01-08

**Summary:**

The initial concerns about the paper were that: it is not sufficiently self-contained about the full agent pipeline, generalization beyond DAGs not necessarily found outside Minecraft, and lack of novelty in the overall method design. That said, the reviewers agree that the underlying problem is important and the empirical results support many of the claims initially made.

**Reviewer Concerns:**

The rebuttal response from the authors was quite extensive and many new experiments were added to address them.

Generalization outside Minecraft: TextWorld cooking, where agents attempt to cook inside a house in a text game setting, were added

Clarity about the design choices were also added which improves the overall readability.

The only real remaining concern is regarding novelty of the underlying method but this can be discounted given the strong empirical results

**Reviewer Scores:**

VSc5 would have increased their score to at least a 6 by their own comment

JH8W stated that their score would increase from 4 to an 8!

DZb7 seems likely to keep their score at 6 given the discussion and their continuing concerns about inference costs

S6YL also will likely keep their score at 4 given they do not seem convinced by the experiments to show generalization outside Minecraft

---

### Decision · Program_Chairs · 2026-01-26

Accept (Poster)